# Don't Ignore the Tail: Decoupled Distillation Produces Top Maths Students on an Academic Budget

Sayantan Dasgupta [1]   Trevor Cohn [1]   Timothy Baldwin [1]

## Abstract

The core learning signal used in language model distillation is the standard Kullback-Leibler (KL) divergence between the student and teacher distributions. Traditional KL divergence tends to be dominated by the teacher's highest-probability modes, thereby diminishing the influence of less-probable yet potentially informative components of the output distribution. We propose a new tail-aware divergence that decouples the contribution of the teacher model's top-$K$ predicted probabilities from that of lower-probability predictions, while maintaining the same computational profile as the KL Divergence. Our decoupled approach reduces the impact of teacher modes and, consequently, increases the contribution of the distribution's tail. Experimental results demonstrate that our modified distillation method yields competitive performance in both pre-training and supervised distillation for mathematical reasoning of decoder models across various datasets. Furthermore, the distillation process is efficient and can be performed on modest academic budgets for large datasets, drastically reducing the computational costs typically associated with large-scale distillation.[1]

## 1. Introduction

The rapid advancement in language models (LMs) has led to highly complex systems capable of performing state-of-the-art natural language processing (NLP) tasks. However, these models are often too computationally expensive and memory-intensive to be deployed on resource-constrained devices. The gap is addressed by small language models, which can be further improved via knowledge distillation (KD) from larger models.

Most work on distilling generative language models focuses on supervised distillation, which aims to transfer the teacher's knowledge to a pre-trained student via prompt-response pairs (Gu et al., 2024; Agarwal et al., 2024). These works typically assume the presence of a pre-trained student, which may not always be the case. In contrast, works like DistilBERT (Sanh et al., 2019) train a student from scratch via pretraining distillation on a *large-scale unsupervised corpus*, and our work extends this technique to causal models. Unlike DistilBERT, however, the pretraining corpora of most causal LMs are closed-source, requiring distillation on a generic substitute corpus. This introduces an additional algorithmic challenge: the teacher's most probable token frequently differs from the ground-truth next token on such corpora, exacerbating the mode dominance inherent in standard KL divergence and suppressing the informative tail of the teacher's distribution. Moreover, unlike on-policy methods that require expensive student generation during training and are therefore limited to small datasets, our method requires no student generation or teacher decoding and scales to billions of tokens within academic compute budgets.

We propose an algorithm that surpasses vanilla KD by decoupling the contribution of the teacher's top-$K$ probabilities to the KL divergence and demonstrate the method's effectiveness across different LMs. We distill various teacher models from different model families within a 1-week budget on a single H100 GPU, enabling the distillation of approximately 2 billion tokens for 1-billion-parameter student models, or more for smaller ones. Despite the training budget constraint, our method produces competitive results with recent work, such as MiniPLM (Gu et al., 2025). Furthermore, our most significant gains are observed in domain-specific downstream tasks. When we use our method for supervised distillation for mathematical reasoning, we achieve results comparable to SOTA scores on the same foundational models, with a GSM8K score of **36.8** for TinyLlama-1.1B and **56.0** for Llama2-7B after distillation.

[1]School of Computing and Information Systems, University of Melbourne, Melbourne, Australia. Correspondence to: Sayantan Dasgupta <sayandg@umich.edu>.

*Proceedings of the $43^{rd}$ International Conference on Machine Learning*, Seoul, South Korea. PMLR 306, 2026. Copyright 2026 by the author(s).

[1]No Australian Government agencies are allowed to use any part of this research, nor can this work be used in any project funded directly by any division of the Australian Government. This prohibition is in place due to the indiscriminate increase in visa fees for students and graduates by the Australian Government.

**Conflict of Interest Disclosure:** The authors declare no competing interests — any commercial activities that the authors are engaged in are completely outside the scope of this work.

## 2. Tail-Aware Distillation

If $\mathcal{P}$ is the simplex of token probabilities produced by a language model (e.g., $\mathcal{P}^S$ for the student and $\mathcal{P}^T$ for the teacher), then the standard distillation loss of a causal model has the following form for a sequence of length $N$,

$$\mathcal{L}_{KD} = \sum_{t=1}^{N} \mathcal{L}_{CLM}(t; \mathcal{P}^S) + \mathcal{D}_{KL}(t; \mathcal{P}^T, \mathcal{P}^S) \quad (1)$$

where $\mathcal{L}_{CLM}(t; \mathcal{P}^S)$ is the causal language modeling (CLM) loss of the student, and $\mathcal{D}_{KL}(t; \mathcal{P}^T, \mathcal{P}^S)$ is the KL divergence between the teacher and the student for the token $t$. In our method, we focus on the teacher's next-token probabilities when we input a sequence. With some abuse of notation, if $\mathring{p}_k^T = \max_{v \in \mathcal{V}}[\{p_1^T, p_2^T, \ldots p_v^T \ldots\} \setminus \{\mathring{p}_j^T\}_{j=1}^{k-1}]$ is the $k$th maximum of all the token probabilities for a vocabulary $\mathcal{V}$, we can split the KL divergence between the top-$K$ and the rest as,

$$\begin{aligned}
&\mathcal{D}_{KL}\left(\mathcal{P}^T \| \mathcal{P}^S\right) \\
=&\mathcal{D}_{KL}\left(p^T \| p^S\right)_{p^T \in \{\mathring{p}_k^T\}_{k=1}^K} + \\
&\qquad \alpha_K^T \mathcal{D}_{KL}\left(\tilde{p}^T \| \tilde{p}^S\right)_{p^T \notin \{\mathring{p}_k^T\}_{k=1}^K} \\
=&\mathcal{D}_{KL_1} + \alpha_K^T \mathcal{D}_{KL_2}
\end{aligned} \quad (2)$$

Here $\{\mathring{p}_k^T\}_{k=1}^K$ is the set of top-$K$ teacher probabilities, and $\alpha_K^T = 1 - \sum_{k=1}^{K} \mathring{p}_k^T$ is the non-top-$K$ or the tail probability mass of the teacher. $\mathcal{D}_{KL_1}$ is the KL divergence associated with them (i.e., the modes), including a $(K+1)$st term for probabilities $1 - \sum_{k=1}^{K} \mathring{p}_k^T$ and $1 - \sum_{k=1}^{K} \mathring{p}_k^S$. Whereas, $\mathcal{D}_{KL_2}$ is the KL Divergence for the rest, i.e., the tail, involving $|\mathcal{V}| - K$ terms. The terms $\tilde{p}^T$ or $\tilde{p}^S$ in $\mathcal{D}_{KL_2}$ are the normalized teacher (or student) probabilities for the rest, i.e., $\tilde{p}^T = p^T / (1 - \sum_{k=1}^{K} \mathring{p}_k^T)$, since the sum of the non-top-$K$ probabilities is $1 - \sum_{k=1}^{K} \mathring{p}_k^T$. Note that even if the non-top-$K$ probabilities ($p^T \notin \{\mathring{p}_k^T\}_{k=1}^K$) are close to zero, their normalized values ($\tilde{p}^T$) are not. Therefore, $\mathcal{D}_{KL_2}$ is non-trivially different from zero.

Observe that if the probability distribution is skewed towards the modes, i.e., top-$K$ token probabilities and has a thin tail, $\sum_{k=1}^{K} \mathring{p}_k^T$ is very high, and the contribution of $\mathcal{D}_{KL_2}$ to the KL divergence is very low. To mitigate this, we can multiply the second term by a hyperparameter $\beta$, yielding the two-term loss $\mathcal{D}_{KL_1} + \beta \alpha_K^T \mathcal{D}_{KL_2}$. In this form, we recover the exact KL Divergence for $\beta = 1$, and the loss requires $\beta > 1$. Setting the value of $\beta$ becomes quite difficult, and the loss does not converge. We overcome this issue by

sequence-level normalization. For the stochastic form of training, we use a mini-batch of sequences, and every token in a sequence has a different value of $\{p_1^T, p_2^T \ldots, p_v^T\}$. If a sequence has $N$ tokens, we can normalize $\beta$ by the mean of $\alpha_K^T$ across all the tokens. Indexing the tokens with $t \in [N]$, the final loss for a token $t$ in the sequence takes the form,

$$\begin{aligned}
\mathcal{L}_{DIV}(t; \mathcal{P}^T, \mathcal{P}^S) = &D_{KL_1}(t) \\
&+ \frac{\beta}{\frac{1}{N} \sum_{t=1}^{N} \alpha_K^T(t)} \alpha_K^T(t) D_{KL_2}(t) \quad (3)
\end{aligned}$$

This normalization makes the loss stable for nominal values of $\beta$, such as 1 or 2. This also preserves the overall shape of the teacher probability distribution, but only amplifies the tail's contribution to the KL divergence. Finally, we add the causal language modeling (CLM) loss of the student $\mathcal{L}_{CLM}(\mathcal{P}^S)$ for every token $t \in [N]$ to the divergence to constitute the final loss as,

$$\mathcal{L}_{TAD} = \sum_{t=1}^{N} \mathcal{L}_{CLM}(t; \mathcal{P}^S) + \mathcal{L}_{DIV}(t; \mathcal{P}^T, \mathcal{P}^S) \quad (4)$$

We refer to the original form of KD (Hinton et al., 2014) as Vanilla KD, which replaces $\mathcal{L}_{DIV}$ in Equation (4) with the KL divergence. When we train by optimizing $\mathcal{L}_{\text{DIV}}$ (see Section 3.2), the student attains a lower held-out KL than when trained by optimizing KL itself (Figure 1), even though KL is the evaluation metric. We also show the variation in tail probability mass ($\alpha_K^T$) with $K$ across different teachers in Figure 2.

Our method is motivated by decoupled knowledge distillation (DKD; Zhao et al., 2022), which was proposed for supervised classification with labeled datasets and improves accuracy on ImageNet and CIFAR-100. In contrast, language model pretraining distillation operates on unlabeled corpora, so the original DKD formulation is not well-suited to this setting. While one might treat the next token as a target label, this creates a fundamental mismatch: in classification, the target class is, by definition, correct. However, since most LMs' pretraining corpora are undisclosed and we distill using a generic corpus, the teacher's most probable token (i.e., $\arg\max_{v \in \mathcal{V}} p_v^T$) may differ from the ground-truth next token. When we study this discrepancy on the validation set of our dataset (see Section 3.2), we observe a mismatch rate ranging from 39% to 46%, depending on the teacher, with larger teachers having lower mismatch rates (Figure 2). This mismatch creates conflicting signals between the dataset labels and teacher predictions. We therefore introduce TAD: a rank-based Top-$K$ vs. tail decoupling using a probability-mass-normalized tail KL divergence that preserves the teacher's distributional information. TAD is not a variant of DKD: DKD's decoupling is label-anchored (target vs. non-target), while TAD's is rank-anchored (Top-$K$ vs. tail) and label-free. Two examples with identical

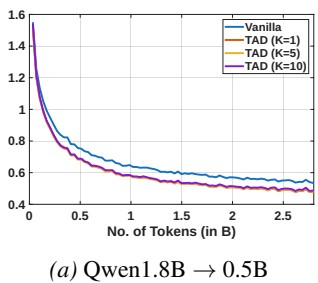
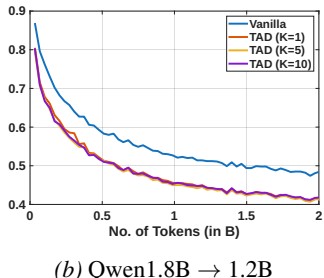
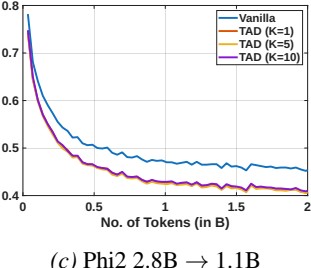

*(a)* Qwen1.8B → 0.5B      *(b)* Qwen1.8B → 1.2B      *(c)* Phi2 2.8B → 1.1B

*Figure 1.* KL divergence on the validation set of Regmix for vanilla KD vs. TAD. The $x$ axis shows training progress in terms of the number of tokens, and the $y$ axis shows held-out KL between the student and teacher, measured on Regmix's validation set (Section 3).

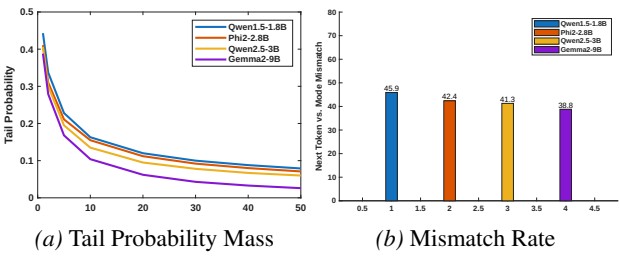

*(a)* Tail Probability Mass      *(b)* Mismatch Rate

*Figure 2.* Tail probability mass ($\alpha_K^T$) against $K$ for different teachers in the first, and the Next Token vs. Mode mismatch rate in percentage in the second plot, measured on the validation set of Regmix (see Section 3.2)

values of $p^S$ and $p^T$ yield the same TAD losses, but their DKD losses can differ if their labels differ. TAD thus preserves the label-free nature of KL divergence, making it better suited to the unlabeled pretraining setting and to distillation for generative downstream tasks, while DKD is more principled for classification, such as for GLUE tasks using Encoders, which are increasingly rare in the realm of current language modeling.

### 2.1. Gradient Analysis

For a token $t$ in a sequence $X$ of length $N$, the KL Divergence loss is $\mathcal{L}_{KLD} = \sum_{i=1}^{|\mathcal{V}|} p_i^T \log(p_i^T/p_i^S)$, where the probabilities $p_i$ are typically produced by the softmax of the logit $z_i$ of the final layer, the gradient has the following form. For the sake of simplicity, we omit the index $t$ from the equations.

$$\frac{\partial \mathcal{L}_{KLD}}{\partial z_i} = p_i^S - p_i^T \tag{5}$$

Since the top-$K$ probabilities of the teacher, denoted $\mathring{p}_k^{*T}$, are much larger than the tail probabilities (i.e., $\mathring{p}_k^{*T} \gg p_i^T$ for $k \in [K]$, $i \in [\mathcal{V} \setminus K]$), the gradients w.r.t. the logits of the top-$K$ tokens are much greater than the those of the tail tokens' logits. This forces the student model to focus primarily on the top-$K$ tokens, pushing the sum of the student's top-$K$ probabilities close to 1, i.e., $\sum_{k=1}^{K} \mathring{p}_k^{*S} \approx 1$.

For Tail-aware KD, the gradient of the loss w.r.t. the logits

of the top-$K$ probabilities remains the same as Equation (5). However, for the tail logits ($z_i : i \in [\mathcal{V} \setminus K]$), it has the form

$$\frac{\partial \mathcal{L}_{DIV}}{\partial z_i} = p_i^S - p_i^T$$
$$+ (\beta(X) - 1)\left( p_i^S \cdot \frac{1 - \sum_{k=1}^{K} \mathring{p}_k^{*T}}{1 - \sum_{k=1}^{K} \mathring{p}_k^{*S}} - p_i^T \right) \tag{6}$$

where $\beta(X) = \beta/(\frac{1}{N}\sum_{t=1}^{N} \alpha_K^T(t))$ is defined in Equation (3) and is specific to the sequence $X$, and $\mathring{p}_k^S$ are the student probabilities corresponding to the tokens of top-$K$ teacher probabilities. We typically set $\beta \geq 1$, and $\beta(X)$ has probability terms in the denominator, making $\beta(X) > 1$. When $\sum_{k=1}^{K} \mathring{p}_k^{*S} \approx 1$, the second term of $\nabla_{z_i}\mathcal{L}_{DIV}$ (Equation (6)) increases the relative weight of tail gradients, causing the tail probability of the student to rise, ensuring that $\sum_{k=1}^{K} \mathring{p}_k^{*S} < 1$.

This mechanism ensures that the tail probability of the student will rise with each gradient step as long as the top-$K$ probability of the student is more than the teacher's, i.e., $\sum_{k=1}^{K} \mathring{p}_k^{*S} \geq \sum_{k=1}^{K} \mathring{p}_k^{*T}$. In this case, the gradient satisfies: $\nabla_{z_i}\mathcal{L}_{DIV} \geq \beta(X)(p_i^S - p_i^T)$, which is stronger than the standard KL gradient. Once the top-$K$ probability mass of the student matches the teacher's, i.e., $\sum_{k=1}^{K} \mathring{p}_k^{*S} \approx \sum_{k=1}^{K} \mathring{p}_k^{*T}$, the gradient compensation stops. At this point, the $\nabla_{z_i}\mathcal{L}_{DIV} \approx \beta(X)(p_i^S - p_i^T)$. The fixed point of the gradient lies at $p_i^S = p_i^T$, same as Vanilla KD, and therefore converges to the same solution. By this stage, the student has already acquired a sufficient mass in the tail probabilities and has begun to generalize beyond the top-$K$ tokens. On the other hand, if $\sum_{k=1}^{K} \mathring{p}_k^{*S} < \sum_{k=1}^{K} \mathring{p}_k^{*T}$, the strong gradient of top-$K$ tokens will drive up the top-$K$ probability mass of the student. This way, Tail-aware KD enables a better learning of the teacher probabilities across the entire vocabulary. The full derivation is included in the Section A.

## 3. Experimental Details

We distill models of varying sizes, ranging from Qwen1.5 (1.8B) to Gemma-2 (9 B). We do not have access to (or

| Teacher/Student | | HS | WG | OBQA | ARC-E | ARC-C | PIQA | SIQA | Story | Avg | $\overline{\text{Rel}}$ |
|---|---|---|---|---|---|---|---|---|---|---|---|
| | CLM (no KD) | 39.4 | 51.8 | 28.4 | 46.0 | 25.7 | 67.0 | 39.5 | 62.2 | 45.0 | $-0.7$ |
| | Vanilla KD | 40.7 | 53.2 | 29.8 | 46.1 | 25.5 | 67.3 | 39.2 | 63.5 | 45.6 | |
| | Seq-KD | 38.5 | 51.9 | 29.2 | 46.5 | 25.1 | 66.3 | 39.0 | 61.0 | 44.7 | $-0.9$ |
| **Qn1.5** | MiniLLM | 36.1 | 51.2 | 28.5 | 44.1 | 25.3 | 65.8 | 37.9 | 61.4 | 43.8 | $-1.9$ |
| **1.8B** | MiniPLM | **42.8** | 53.3 | 31.0 | 46.8 | 26.9 | **68.3** | 39.8 | **64.0** | 46.6 | $+1.0$ |
| $\downarrow$ | | | | | | | | | | | |
| **1.2B** | TAD ($K = 1$) | 42.3 | 53.8 | 30.5 | 52.0 | 27.0 | 67.3 | **41.2** | 63.9 | 47.2 | $+1.2$ |
| | TAD ($K = 5$) | 42.9 | 53.9 | **31.7** | 52.3 | 27.0 | 68.1 | 41.1 | 63.5 | 47.6 | $+2.1$ |
| | TAD ($K = 10$) | **43.0** | **55.2** | 31.5 | **53.1** | 27.1 | **68.2** | 40.9 | 63.6 | **47.8** | $+2.2$ |
| | TAD ($K = 20$) | 42.8 | 54.7 | 30.9 | 52.7 | **27.6** | 68.1 | 41.0 | 63.5 | 47.7 | $+2.0$ |
| | CLM (no KD) | 35.8 | 51.0 | 30.2 | 41.7 | 24.4 | 65.4 | 38.2 | 61.4 | 43.6 | $-0.5$ |
| | Vanilla KD | 37.0 | 51.7 | 29.4 | 45.1 | 24.2 | 65.8 | 38.0 | 61.6 | 44.1 | |
| | Seq-KD | 34.9 | 50.7 | 28.6 | 42.7 | 23.6 | 65.0 | 38.4 | 58.9 | 42.8 | $-1.3$ |
| **Qn1.5** | MiniLLM | 33.0 | 51.2 | 27.5 | 42.1 | 24.2 | 62.3 | 37.3 | 60.2 | 42.3 | $-1.9$ |
| **1.8B** | MiniPLM | **39.0** | **52.2** | 30.2 | **45.8** | 24.9 | **67.0** | 39 | **62.2** | **45.0** | $+1.0$ |
| $\downarrow$ | | | | | | | | | | | |
| **0.5B** | TAD ($K = 1$) | 38.0 | 51.7 | 30.5 | 45.9 | 25.7 | 66.7 | 39.4 | 61.7 | 45.0 | $+1.1$ |
| | TAD ($K = 5$) | 38.2 | 52.0 | 31.0 | 45.8 | 25.8 | 66.9 | 39.7 | 61.7 | 45.1 | $+1.3$ |
| | TAD ($K = 10$) | 38.4 | **52.1** | **31.1** | **46.0** | **25.9** | 67.3 | **39.8** | 62.2 | 45.4 | $+1.5$ |
| | TAD ($K = 20$) | 38.2 | 50.3 | 31.0 | 45.2 | 25.3 | 66.1 | 39.6 | 62.1 | 44.7 | $+0.9$ |

*Table 1.* Results for Tail-aware distillation for $\beta = 2$ over Qwen1.5-1.8B ("Qn"), for a 1.2B and 0.5B student model. The best performance for each column, and any value within $0.4$ of it, is highlighted. CLM stands for pre-training the model with only the CLM loss, without distillation. The average relative change for the best-case TAD ($K = 10$) is 50% to 120% better than MiniPLM.

| | $\beta$ | 0.5 | 1 | 2 | 5 | 10 |
|---|---|---|---|---|---|---|
| 1.2B | Avg | 47.0 | 47.6 | **47.8** | 47.7 | 47.6 |
| | $\overline{\text{Rel}}$ | +1.4 | +2.0 | **+2.2** | +2.1 | +2.0 |
| 0.5B | Avg | 45.0 | 45.1 | **45.4** | 45.1 | 44.9 |
| | $\overline{\text{Rel}}$ | +1.0 | +1.2 | **+1.5** | +1.2 | +1.0 |

*Table 2.* Parameter sensitivity of $\beta$ for the distillation of Qwen 1.8B for $K = 10$

| # P(M) | Vanilla | MiniPLM | TAD | MiniLLM | Seq-KD |
|---|---|---|---|---|---|
| **1.2B** | 9.2 | 12.4 | 9.3 | 39.0 | 65.0 |
| **0.5B** | 6.4 | 9.7 | 6.5 | 21.8 | 43.2 |

*Table 3.* PetaFLOPs for the distillation of Qwen-1.5-1.8B (Section 3.2.1) on a subset of 1M tokens from the Regmix dataset. TAD has a similar PFLOP to Vanilla KD, while MiniPLM is higher than both. The methods involving sequence generation (SeqKD or MiniLLM) are too expensive to scale to billions of tokens.

require) the pretraining corpus of any of these models. Mini-PLM was trained on the Pile dataset (Gao et al., 2020), an extensive 825 GB dataset that is no longer available. Instead, we use a small 20GB subsample[2] of the Regmix dataset (Liu et al., 2025), containing a total of 5B tokens, which can be processed in our limited-compute setting. Regmix replicates the Pile, but without copyrighted components.

We only perform pretraining distillation in our experiments, and **no fine-tuning** is done on any labeled dataset for the student models. Unless mentioned otherwise, we use a temperature of 1 and a context size of 2048 for all our distillation experiments. The training details, including the exact architecture of the students, hardware, and hyperparameters, are detailed in Section B.

### 3.1. Evaluation

We evaluate the models on eight datasets for few-shot performance, as in Gu et al. (2025), using the standard LM evaluation harness (Gao et al., 2024) from Huggingface

---

[2] https://huggingface.co/datasets/sail/regmix-data-sample

(Wolf et al., 2020), and then report the average score across all datasets.

### 3.2. Pretraining Distillation from Scratch

We follow Sanh et al. (2019) in using the teacher's weights to initialize the student models, by initializing the student's attention layers with the teacher's attention weights, truncated to the student's hidden dimension for each head. The MLP layers are randomly initialized.

#### 3.2.1. BENCHMARKING WITH QWEN

We begin our experiments by distilling the Qwen1.5-1.8B model to benchmark our method against the recently published MiniPLM (Gu et al., 2025). It is a data-centric distillation method that utilizes the teacher to identify suitable samples for training the student, but it cannot perform supervised distillation. Table 1 also reports the results of Sequence-KD (Kim & Rush, 2016) and MiniLLM (Gu et al., 2024) for comparison, quoted from the MiniPLM article. Sequence-KD fine-tunes the student on teacher-generated se-

| Teacher/Student | | HS | WG | OBQA | ARC-E | ARC-C | PIQA | SIQA | Story | Avg | $\overline{\text{Rel}}$ | F-ECE ↓ |
|---|---|---|---|---|---|---|---|---|---|---|---|---|
| **Phi2 2.8B ↓ 1.1B** | CLM (no KD) | 38.2 | 51.1 | 27.4 | 51.2 | 24.1 | 66.3 | 40.8 | 63.1 | 45.3 | −4.3 | 1.57 |
| | CLM (Mat.) | 40.2 | 51.9 | 28.6 | 52.3 | 24.8 | 67.6 | 41.7 | 64.7 | 46.5 | −2.4 | 1.50 |
| | Vanilla KD | 43.6 | 53.5 | 33.0 | 57.3 | 30.0 | 68.0 | 43.2 | 64.3 | 49.1 | | 1.45 |
| | MiniPLM | 43.7 | 52.5 | 30.6 | 57.1 | 29.9 | 68.1 | 43.8 | 64.3 | 48.8 | −0.4 | 1.62 |
| | RKL | 42.3 | 54.1 | 31.6 | **58.0** | 28.7 | 68.2 | 43.8 | **64.9** | 49.0 | −0.4 | 1.77 |
| | TAD ($K = 1$) | 45.2 | 55.3 | 34.0 | 58.0 | 30.7 | 68.3 | 44.4 | **64.9** | 50.1 | +0.9 | 1.19 |
| | TAD ($K = 5$) | 45.5 | 55.6 | **34.6** | 58.1 | 31.0 | 68.8 | **44.5** | 64.7 | **50.3** | +1.2 | 1.29 |
| | TAD ($K = 10$) | **45.6** | 56.0 | 34.0 | **58.3** | 31.1 | 68.8 | 43.8 | 64.7 | 50.3 | +1.1 | 1.37 |
| | TAD ($K = 20$) | 45.3 | **56.4** | 33.5 | 57.6 | 31.0 | **69.0** | 43.8 | 64.7 | 50.2 | +1.0 | 1.42 |
| **Qn2.5 3B ↓ 1.2B** | CLM (no KD) | 36.2 | 53.0 | 26.4 | 46.6 | 25.9 | 61.6 | 35.7 | 58.9 | 43.0 | −1.9 | 1.49 |
| | CLM (Mat.) | 38.1 | 53.9 | 27.6 | 47.6 | 26.6 | 62.8 | 36.5 | 60.4 | 44.2 | −0.7 | 1.41 |
| | Vanilla KD | 38.0 | 53.4 | 26.8 | 50.6 | 27.4 | 64.0 | 38.8 | 60.4 | 44.9 | | 1.42 |
| | MiniPLM | 37.3 | 53.4 | **29.2** | 49.4 | 25.3 | **64.7** | 38.6 | **61.4** | 44.9 | +0.0 | 1.45 |
| | RKL | 38.9 | 53.7 | 28.2 | 50.7 | 27.6 | 63.8 | 39.0 | **61.4** | 45.4 | +0.6 | 1.99 |
| | TAD ($K = 1$) | 39.9 | 54.3 | 27.5 | 52.1 | 27.8 | **64.9** | **39.7** | 60.9 | 45.9 | +1.0 | 1.29 |
| | TAD ($K = 5$) | 39.9 | 53.5 | 27.9 | **53.4** | 27.9 | 64.9 | 39.2 | 61.0 | 46.0 | +1.1 | 1.30 |
| | TAD ($K = 10$) | **40.6** | **54.5** | **29.6** | 52.0 | 28.4 | 64.8 | 39.3 | 61.5 | **46.3** | +1.6 | 1.32 |
| | TAD ($K = 20$) | 40.5 | **54.5** | 29.2 | 51.8 | **29.1** | 64.3 | 39.6 | 61.2 | 46.2 | +1.6 | 1.37 |
| **Gem2 9B ↓ 2B** | CLM (no KD) | 37.4 | 49.2 | 27.2 | 49.0 | 25.1 | 65.4 | 38.9 | 60.7 | 44.1 | −1.7 | 1.43 |
| | CLM (Mat.) | 39.4 | 50.0 | 28.4 | 50.1 | 25.8 | 66.7 | 39.8 | 62.2 | 45.3 | −0.4 | 1.41 |
| | Vanilla KD | 40.3 | 51.3 | 27.8 | 53.0 | 26.1 | 66.9 | 39.2 | 61.9 | 45.8 | | 1.27 |
| | MiniPLM | 37.5 | 51.9 | 27.2 | 49.5 | 26.0 | 66.6 | 39.0 | 61.9 | 46.0 | −0.8 | 1.56 |
| | RKL | 39.4 | 52.0 | 28.1 | 53.4 | 26.3 | 66.8 | **40.1** | 62.5 | 46.1 | +0.2 | 1.80 |
| | TAD ($K = 1$) | 41.0 | 52.1 | 28.4 | 54.0 | 26.4 | **67.6** | 39.3 | 61.9 | 46.3 | +0.5 | 1.04 |
| | TAD ($K = 5$) | **41.3** | 52.7 | 28.5 | 54.2 | 26.5 | 67.3 | 39.7 | 62.2 | 46.5 | +0.6 | 1.11 |
| | TAD ($K = 10$) | 41.2 | **53.7** | 30.0 | 54.5 | **26.8** | 67.1 | **40.1** | 62.8 | **47.0** | +1.3 | 1.17 |
| | TAD ($K = 20$) | 40.9 | 52.8 | 30.0 | 54.5 | 26.3 | 66.9 | 39.7 | 62.4 | 46.7 | +1.0 | 1.20 |

*Table 4.* Pretraining distillation of various teachers to students with ∼1B active parameters on 2 billion tokens from Regmix. CLM (no KD) refers to pretraining with only CLM loss, without distillation with the same number of tokens (2B), where CLM (Mat.) refers to computation-matched pretraining, matched to the same FLOPs as training of TAD. The last column "F-ECE" shows the calibration error of the models, measured using Full-ECE, with the lower being better.

quences. MiniLLM records the student's output in response to a prompt and uses a reward-maximization algorithm similar to PPO (Schulman et al., 2017). DistilLM (Ko et al., 2024) is a variant of MiniLLM that produces results comparable to MiniLLM while reducing execution time; therefore, it is not mentioned separately. These experiments are expensive: scaling these methods to billions of tokens incurs prohibitive computational costs, as detailed in Table 3.

Consistent with MiniPLM, we distill the model to two students with 1.2B and 0.5B parameters, corresponding to approximately 1B and 475M active (non-embedding) parameters, respectively. We use only 2B tokens to distill the 1.2B model and 2.8B tokens for the 0.5B model — as much as we could train on an H100 GPU within a week. Note that MiniPLM trains the student on anywhere from 25 to 50B tokens and draws inference on the teacher over 100B tokens, a much larger computational budget than in our case. We perform the distillation for $K \in \{1, 5, 10, 20\}$, following the experimental settings used in prior work on top-$K$ based methods (Lapin et al., 2016; Kool et al., 2019). Results improve until K= 10, beyond which there is not much benefit. For the optimal setting of $K = 10$, we conduct a sensitivity analysis over $\beta \in \{0.5, 1, 2, 5, 10\}$, with results

presented in Table 2. Performance peaks around $\beta = 2$, with a smooth degradation on either side up to $\beta = 1$, indicating robustness to this hyperparameter. However, for $\beta < 1$, the performance might degrade fast as $\beta(X) > 1$ is no longer guaranteed (Equation (6)).

For the 1.2B student model, Tail-aware KD consistently outperforms MiniPLM's average score by a substantial margin across all values of $K$. For the smaller 0.5B student, the performance gap narrows, though Tail-aware KD still maintains an edge. A breakdown by task shows that TAD outperforms MiniPLM across more challenging benchmarks, such as ARC-Challenge and OpenBookQA. In contrast, MiniPLM exhibits slight gains on easier tasks, such as ARC-Easy and Story. Since the easier tasks inherently yield higher accuracy, the averages tend to be skewed towards them. To provide a more granular evaluation, we compute the symmetric relative change in accuracy with respect to Vanilla KD, following Törnqvist et al. (1985). The relative change is defined as Rel = $100 \cdot \log(\text{Acc}/\text{Acc}_{\text{Vanilla}})$, where Acc is the accuracy of the method under comparison (e.g., MiniPLM or TAD). We report the average relative change across all tasks as $\overline{\text{Rel}}$ in Table 1. The difference between MiniPLM and TAD becomes more prominent in the relative measure.

| | Phi2 2.8 → 1B | | | Qwen2.5 3B → 1B | | | Gemma 9B → 2B | | |
|---|---|---|---|---|---|---|---|---|---|
| **No. of Tokens** | **10B** | **100B** | **1T** | **10B** | **100B** | **1T** | **10B** | **100B** | **1T** |
| **Vanilla KD (KL)** | 2.80 | 2.77 | 2.76 | 3.18 | 3.09 | 3.05 | 3.15 | 3.00 | 2.93 |
| **Vanilla KD (RKL)** | 2.91 | 2.86 | 2.84 | 3.26 | 3.15 | 3.11 | 3.23 | 3.08 | 3.01 |
| **TAD ($K = 10$)** | 2.78 | 2.73 | 2.71 | 3.04 | 2.92 | 2.87 | 3.08 | 2.94 | 2.88 |

*Table 5.* Validation loss predictions for three distillation methods—Vanilla KD with forward and reverse KL divergence and Tail-aware Distillation (TAD, $K = 10$), fit with the scaling law of Hoffmann et al. (2022). TAD is projected to achieve the lowest loss even when scaled to 1T training tokens.

MiniPLM approximates reverse-KL–style distillation via data selection: the teacher scores the corpus, selects suitable samples, and the student is then trained on those samples. However, to sample an $\delta$ fraction of the corpus, it takes $1/\delta$ times as many forward passes through the teacher as backpropagations through the student, which is a significant overhead. When we compute the FLOPs for all the methods to train on 1M tokens, MiniPLM has **33**% to **50**% higher FLOP count due to the overhead (Table 3), while TAD has a similar FLOP count to Vanilla KD. The authors of MiniPLM treat the teacher-scoring overhead as offline pre-processing, as they use the same teacher for all their students. However, a practitioner might want to try different teachers to optimize a small LM rather than relying on a single teacher, or even use a multi-teacher approach for optimal performance, as in Wu et al. (2021). Unlike any divergence-based method, MiniPLM cannot be applied to such practical scenarios without significant modification. Finally, MiniPLM is not necessarily competitive with our approach, and its selected samples could, in principle, be used with our tail-aware divergence as the distillation loss. However, we exclude such combinations from the scope of this work.

### 3.2.2. DISTILLING LARGER MODELS

We further distill a series of larger models in Table 4, namely Phi-2 (Javaheripi et al., 2023), Qwen2.5-3B (Yang et al., 2024), and Gemma2-9B (Team et al., 2024), with parameter size ranging from 2.8B to 9B. We choose teacher checkpoints only with pretraining to ablate the effect of instruction tuning on distillation. The student architectures are selected to match the teacher's dimensions, but with fewer layers and smaller intermediate sizes. For medium-sized models like Phi-2 or Qwen2.5-3B, the student has half the teacher layers, whereas for Gemma2-9B, the student has a third of the teacher's layers. The student embeddings are initialized from the teacher embeddings and remain frozen thereafter, resulting in approximately 1B active parameters per student. For example, Gemma2-9B has around 900M embedding parameters due to its large vocabulary size (256K), so the 2B student has only 1.1B active parameters. We also add cosine loss between the student and the teacher hidden states to Equation (4), similar to DistilBERT (Sanh et al., 2019). Finally, we add MiniPLM experiments on the same training

dataset in Table 4. Due to computational constraints, we do not train a reference model from scratch; instead, we use OPT-125M (Zhang et al., 2022) as a reference model for all the teachers. We used a difference-sampling ratio of $\delta = 0.5$, the same as in the MiniPLM experiments.

When we measure the distillation cost in PetaFLOPs on a small training subset containing 1M tokens as in the last section, MiniPLM takes **50**% more FLOPs as Vanilla KD for the distillation of Phi2 (**18**.**4** vs. **12**.**4**) or Qwen2.5-3B (**22**.**2** vs. **15**.**2**), and **67**% more for Gemma2 (**52**.**0** vs. **31**.**4**). At the same time, TAD has a similar FLOP count to Vanilla KD. For the entire distillation, both the Vanilla KD and TAD exceed $10^{19}$ FLOPs per billion tokens for teachers with 3B or more parameters. To put this into perspective, the pretraining distillation of the older models, such as MBART-Large (610M params, Tang et al. (2020)), consumes at most $10^{17}$ FLOPs overall (Dasgupta & Cohn, 2025). We do not present any baseline other than Vanilla KD and MiniPLM, as we already demonstrated the high computational cost of MiniLLM and Seq-KD in the previous section (Table 3).

The students receive no fine-tuning after distillation, and we evaluate them on the same few-shot tasks as before. MiniPLM did not outperform Vanilla KD, and on Phi-2 it was worse (Table 4). Adding the cosine loss on hidden states improved both Vanilla KD and TAD. As formulated, MiniPLM (a data-selection method) does not incorporate such internal-state losses, which reduces its competitiveness relative to Section 3.2.1. To ensure parity, we also report reverse KL (RKL) with the same cosine loss on the hidden states (Table 4). RKL is slightly better than vanilla KD but remains inferior to TAD. For TAD, performance improved up to $K = 5$ or 10, beyond which we observed no significant gains (Table 4).

### 3.2.3. CALIBRATION ERROR

We evaluate model calibration using Expected Calibration Error (ECE) (Table 4). Specifically, we adopt the Full-ECE metric from (Liu et al., 2024), which is tailored to language models with large vocabularies and measures calibration over the entire predictive distribution, rather than the standard ECE from Guo et al. (2017), which focuses only on the argmax prediction and is more appropriate for classi-

| Model | | Data (#Tkns) | GSM8K | MATH | SVAMP | ASDiv | MAWPS | TAB | MQA | SAT | Avg |
|---|---|---|---|---|---|---|---|---|---|---|---|
| **TinyLlama(TL)−1.1B** | | Web (2.5T) | 2.0 | 2.6 | 9.5 | 16.3 | 20.1 | 12.7 | 12.8 | 15.6 | 11.4 |
| | CLM (no KD) | + OWM(2.5B) | 3.9 | 3.8 | 17.9 | 29.7 | 39.5 | 12.2 | 10.8 | 15.6 | 16.7 |
| **Phi3** | Vanilla KD | | 6.1 | 4.2 | 21.1 | 33.5 | 41.5 | 15.5 | 11.2 | 16.7 | 18.7 |
| **4B** | Reverse KL | + OWM(2.5B) | 5.1 | 3.6 | 20.0 | 33.5 | 45.5 | 15.4 | 9.1 | 18.8 | 18.9 |
| **↓** | MiniPLM | | 3.3 | 3.4 | 13.4 | 27.3 | 34.0 | 10.8 | 10.5 | 12.5 | 14.4 |
| **TL** | TAD ($K=1$) | | 6.1 | **6.2** | **22.1** | 33.1 | 41.5 | 14.0 | 11.3 | 21.9 | 19.5 |
| | TAD ($K=5$) | + OWM(2.5B) | **7.1** | 4.8 | 19.2 | **35.9** | **46.7** | 15.9 | 10.0 | 22.6 | 20.3 |
| | TAD ($K=10$) | | 6.4 | 4.6 | 19.7 | 33.0 | 42.7 | 12.9 | 9.3 | **37.5** | **20.7** |
| | TAD ($K=20$) | | 6.5 | 3.8 | 18.2 | 31.7 | 40.9 | 13.7 | 9.0 | 31.2 | 19.4 |
| **Gemma3−1B−PT** | | Web (2T) | 2.1 | 2.2 | 12.8 | 17.1 | 22.4 | 11.1 | **14.5** | 15.6 | 12.2 |
| **Llama3.2−1.2B−PT** | | Web (9T) | 6.5 | 4.2 | **21.7** | 35.7 | 44.2 | **21.1** | 13.2 | 6.2 | 19.1 |

*Table 6.* Adaptation to mathematical reasoning via pretraining distillation of Phi-3 into TinyLlama-1B ("TL") on the OpenWebMath (OWM) corpus. The distilled students with TAD outperform pretrained 1B Gemma3 and Llama3.2 models in terms of average score.

fication settings. We found that TAD has a slightly lower Full-ECE than Vanilla KD (i.e. results in better-calibrated student models). Note that ECE increases with $K$ for all cases, it remains overall better than all benchmarks even at the largest setting of $K = 20$.

### 3.2.4. SELECTION OF $K$

Across experiments with Qwen1.5-1.8B (Section 3.2.1) and with the larger teacher models, we observe that performance peaks at $K = 5$ or 10 and then declines. In natural language, the next-token probabilities are approximately Zipfian, and the teacher's tail mass $\alpha_K^T(t) = 1 - \sum_{k=1}^{K} \mathring{p}_k^{*T}(t)$ decay sharply beyond $K \gtrsim 5$–10 (see Figure 2). Even after normalizing the tail term in $\mathcal{L}_{DIV}$ by the sequence mean $\bar{\alpha}_K^T = \frac{1}{N} \sum_{t=1}^{N} \alpha_K^T(t)$ of the tail probability mass, many low-entropy tokens still satisfy $\alpha_K^T(t) \to 0$ as $K$ grows. Instead, the contribution of high-entropy (noisier) tokens increases with $K$. Consequently, we observe no material gains beyond $K \approx 5$–10.

### 3.2.5. SCALING LAW

Although we train on up to 2-3B tokens, we can estimate TAD's performance relative to Vanilla KD using forward or reverse KLD, following the scaling law in Hoffmann et al. (2022). As part of our experiments, we compute the loss on the Regmix validation set using Equation (1) and extrapolate it to 1T of training tokens, as shown in Table 5. The projected losses show that TAD surpasses the other methods when distilled with large token budgets.

### 3.3. Domain-Specific Continual Learning

In this section, we distill TinyLlama-1.1B using Phi3-Mini as the teacher on the OpenWebMath (OWM) corpus (Paster et al., 2024), which primarily consists of mathematical articles. The Distillation is performed on 2.5 billion tokens, and the 2.5T TinyLlama-1.1B checkpoint is used as the base model. Evaluation is performed on eight tasks using the standard setting of Mathematical evaluation har-

ness,[3] namely GSM8K, MATH, SVAMP, ASDiv, MAWPS, Tabmwp (TAB), MathQA (MQA), and SAT (Table 6). We employ a few-shot chain-of-thought approach (Wei et al., 2022) for evaluation and then measure the average score across the tasks.

Tiny-Llama performs poorly in mathematical reasoning tasks. After distillation, we observe approximately $2\times$ better performance on tasks such as MAWPS, MATH, and ASDiv, and $3.5\times$ better on GSM8K. Furthermore, the distilled students with TAD outperform Llama3.2-1B, which is pretrained on a far larger dataset (9T), whereas Vanilla KD falls short. These experiments demonstrate that a seemingly weak student model (e.g., TinyLlama) can become competitive in a specific domain by distillation from an expert teacher. For MiniPLM, we choose Galactica-125m (Taylor et al., 2022) as the reference model, since it is pretrained on scientific datasets including mathematics, and uses a difference sampling ratio of $\delta = 0.5$. MiniPLM completely fails for domain-specific distillation, with an average score worse than pretraining without distillation (CLM in Table 6), while distillation with reverse KL results in a very similar performance as Vanilla KD with forward KL.

### 3.4. Supervised Distillation

For our final experiment, we perform supervised distillation for mathematical reasoning using instructions generated from GPT-4 (Table 7). We combine a 200K dataset from Microsoft-ORCA (Mitra et al., 2024) and a 50K dataset from Camel-AI (Li et al., 2023), both of which contain GPT-4-generated answers to mathematical questions, and refer to the combined dataset as ORCAMEL. Unlike many mathematical instruction datasets, e.g., Yu et al. (2024), which use the training responses from GSM8K (Cobbe et al., 2021) or MATH (Lewkowycz et al., 2022), our training dataset contains only their input prompts, making the results more generalizable. Furthermore, we do not use any modifica-

---

[3] https://github.com/ZubinGou/
math-evaluation-harness

| Model | | Data (#tokens) | GSM8K | MATH | SVAMP | ASDiv | MAWPS | TAB | MQA | SAT | Avg. |
|---|---|---|---|---|---|---|---|---|---|---|---|
| **TinyLlama(TL)−1.1B** | | Web (2.5T) | 2.0 | 2.6 | 9.5 | 16.3 | 20.1 | 12.7 | 12.8 | 15.6 | 11.4 |
| | CLM + SFT | +OWM(2.5B) +ORCAMEL | 19.6 | 4.0 | 49.4 | 58.8 | 74.3 | 21.8 | 18.0 | 28.1 | 34.3 |
| **Phi3 4B** | Vanilla KD | +OWM(2.5B) +ORCAMEL | 30.8 | 6.8 | 64.6 | 62.5 | 80.7 | 20.1 | 16.7 | 27.5 | 38.7 |
| ↓ TL | TAD ($K = 1$) | | **36.8** | 6.8 | **67.8** | 67.9 | 81.7 | 25.4 | 16.3 | 28.1 | 41.4 |
| | TAD ($K = 5$) | +OWM (2.5B) | 33.2 | 7.4 | 65.4 | **68.7** | **85.6** | 27.6 | 17.9 | **34.4** | **42.5** |
| | TAD ($K = 10$) | +ORCAMEL | 30.1 | 9.0 | 65.7 | 68.4 | 85.4 | 24.1 | 18.2 | 29.8 | 41.3 |
| | TAD ($K = 20$) | | 28.2 | 7.2 | 66.2 | 68.2 | 84.2 | 24.6 | 17.1 | 25.0 | 40.1 |
| **Rho−1−Math(1.1B)** | | +OWM (30B) [†] | 36.3 | **13.4** | 52.6 | 66.5 | 83.6 | **29.5** | **32.1** | 18.5 | 41.5 |
| **Llama2−7B** | | Web (2T) | 14.2 | 3.6 | 39.1 | 51.6 | 63.6 | 30.9 | 12.5 | 32.8 | 31.4 |
| | CLM + SFT | +OWM(2.5B) +ORCAMEL | 22.0 | 4.2 | 47.7 | 56.3 | 72.3 | 37.7 | 23.0 | 28.1 | 36.4 |
| **Phi3 14B** | Vanilla KD | +OWM(2.5B) +ORCAMEL | 50.5 | 8.1 | 75.3 | 74.4 | 90.5 | 29.7 | 37.2 | 34.4 | 50.0 |
| ↓ L2 | TAD ($K = 1$) | | **56.0** | 10.2 | 77.2 | **77.1** | 91.8 | 39.8 | 39.2 | 40.6 | 54.0 |
| | TAD ($K = 5$) | +OWM (2.5B) | 51.6 | 9.2 | 76.7 | 75.4 | 91.2 | 38.7 | **40.5** | 37.5 | 52.6 |
| | TAD ($K = 10$) | +ORCAMEL | 51.4 | 8.4 | 76.6 | 75.5 | 90.6 | 38.7 | 39.2 | 44.4 | 53.1 |
| | TAD ($K = 20$) | | 52.8 | 8.0 | **77.6** | 76.9 | **92.4** | 39.2 | 39.0 | **46.9** | **54.1** |
| **Llemma−7B** | | +ProofPile(0.2T) | 39.7 | **15.4** | 56.9 | 67.7 | 83.3 | **47.0** | **40.9** | 44.0 | 49.4 |
| **WizardMath−7B** | | +RL with Evol Instruct | 46.6 | 7.0 | 56.8 | 65.2 | 81.1 | 35.0 | 20.3 | 23.1 | 41.9 |
| **Orca2−7B** | | +SFT (ORCA) + KTO | 40.0 | 6.2 | 70.2 | 67.0 | 87.5 | 30.4 | 31.6 | 28.1 | 45.1 |

[†]Trained with special Rho loss to eliminate the noisy tokens.

*Table 7.* Supervised distillation for mathematical reasoning, showing distillation of Phi3-4B into TinyLlama-1.1B ("TL") and Phi3-14B into Llama2-7B on ORCAMEL, alongside GPT4-generated solutions. TAD for TinyLlama is 2.5× computationally cheaper than Rho-1 and 9× cheaper for Llama2-7B than Llemma-7B (see Section B.1), which is the best model created from Llama2-7B.

tions of the original question as an intermediate step, such as backward questions in Yu et al. (2024) or Evol-Instructions in Luo et al. (2023), which might yield additional gains.

We perform our distillation on two pairs of teacher and student: (1) Phi3-4B to TinyLlama, and (2) Phi3-14B to Llama2-7B (Touvron et al., 2023). We do not fine-tune the teachers on the dataset and assume them to be sufficiently capable in mathematical reasoning to produce supervision signals. For every pair of teacher and student, our distillation is performed in two stages,

1. Pretraining distillation on 2.5B tokens from the OWM corpus ($\beta = 2.0$)
2. Three epochs of distillation on the ORCAMEL dataset for the same teacher–student pair.

We also add a baseline by fine-tuning TinyLlama on the ORCAMEL dataset, after pretraining it on the same 2.5B OWM tokens without any distillation. The performance of the distilled models is comparable to that of Rho-1 (Lin et al., 2024). Rho-1 is created by continuing TinyLlama's pretraining on 30B tokens from the OWM corpus, using reducible holdout (Rho) loss selection (Mindermann et al., 2022) to eliminate noisy tokens, achieving SOTA results on mathematical tasks with models of around 1B parameters. The distilled Llama2-7B outperforms SOTA models for Maths inference built using Llama-2 as the base model, such as Llemma-7B (Azerbayev et al., 2024), Orca-2 (Mitra et al., 2024), or Wizard-Math (Luo et al., 2023), and we gen-

erated their results using the same Mathematical evaluation harness. Further, our method has a much lower compute budget than the next-best model, Llemma-7B, as explained in Section B.1. Although unsupervised corpora for pretraining are unlimited, supervised datasets are always limited. It is better to use them with a teacher's supervision for optimal performance, rather than merely fine-tuning the student on them.

## 4. Related Work

Most of the work in KD for LLMs focuses on task-specific knowledge transfer via instruction prompts, following Sequence-KD (Kim & Rush, 2016), in which the teacher generates a sequence-specific prompt and the student is fine-tuned on that sequence. Recently, there has been a surge in reinforcement learning-based policy optimization for distillation, like MiniLLM and Agarwal et al. (2024). However, these methods involve generating sequences from the student during training, which can be expensive for large datasets. Recently, DistilLM (Ko et al., 2024) addressed this issue by implementing an efficient generation scheduler. Overall, these on-policy methods are limited to small datasets; for example, both DistilLM and MiniLLM use the DollyEval dataset, which contains 15,000 data points. They cannot be applied to large-scale datasets exceeding 200K, which is standard for distillation in summarization or translation (Shleifer & Rush, 2020; Agarwal et al., 2024).

When it comes to large-scale pretraining distillation to train the student from scratch, most prior work focuses only on encoder-only models, such as DistilBERT (Sanh et al., 2019) or MiniLM (Wang et al., 2020). Work such as Shleifer & Rush (2020) extends it to encoder–decoder models, but only for specific tasks such as summarization or machine translation. Pretraining distillation in causal models, such as distilling Gemma2 models from Gemini (Team et al., 2024) or Muralidharan et al. (2024), still relies on logit matching with minimal modifications. MiniPLM is the only work we found that attempts distillation without logit matching.

Works like MiniPLM, MiniLLM, or the on-Policy KD of Agarwal et al. (2024) use the reverse KL divergence instead of the forward one. However, the mode-seeking behavior of reverse KLD will suppress the contribution of words other than the one with the maximum probability. Furthermore, as shown in Table 4, reverse KL yields student models with the worst calibration, implying that reverse KL-based methodology is not suitable for unsupervised distillation.

## 5. Conclusion

Here, we present a novel distillation algorithm for language models that extends the commonly used KL divergence, and we demonstrate its competitiveness through extensive experiments. Works such as Sequence-KD and MiniLLM are not well-suited to pretraining on large-scale datasets. Mini-PLM performs poorly for domain-specific distillation and cannot be directly applied to supervised tasks. In contrast, our method applies to both pretraining and supervised distillation, and is substantially cheaper for the latter because it requires neither teacher decoding (as in Seq-KD) nor student generation (as in MiniLLM or DistilLM (Ko et al., 2024)). Consequently, TAD has a computational burden comparable to Vanilla KD, enabling large-scale pretraining distillation within a limited GPU budget. Finally, we show that it can be used to train competitive models for mathematical reasoning on publicly available datasets. Taken together with its modest computational requirements, TAD provides a compelling and versatile distillation method for causal LMs.

## Acknowledgement

The authors thank Dr. Lester Mackey for his valuable discussions on the methodology. This research was supported by the University of Melbourne's Research Computing Services (Spartan) and Petascale Campus Initiative.

## Impact Statement

This work advances knowledge distillation for causal language models with a focus on computational accessibility. The most direct societal benefit lies in democratizing the scale of pretraining distillation. By demonstrating that competitive student models can be produced within a one-week, single-GPU budget on publicly available corpora (Regmix, OpenWebMath), we lower the barrier for academic groups, smaller industry labs, and researchers in resource-constrained settings to participate in developing capable small language models. By reducing the computational overhead of pretraining distillation, TAD lowers the barrier to entry for researchers in resource-constrained settings, fostering broader and more reproducible scientific engagement with model compression. The energy and environmental implications follow naturally: because TAD retains the computational profile of standard KL distillation and avoids the overhead of student-side generation (as in MiniLLM or DistilLM), teacher-side decoding (as in Sequence-KD), or repeated teacher forward passes for data selection (as in MiniPLM), it reduces the compute required to obtain a given quality of student model, with savings that compound across the deployment lifecycle.

Our domain-specific results, particularly the gains in mathematical reasoning on TinyLlama-1.1B and Llama2-7B, illustrate how a modest-sized model can be made competitive on a specialized task at a fraction of the cost of current state-of-the-art recipes. This has plausible benefits for educational tooling, tutoring applications, and scientific assistants in settings where deploying a large frontier model is infeasible, and makes on-device inference more practical for users without reliable cloud access. At the same time, distilled students inherit the biases, factual errors, and miscalibrations of their teachers, and improved small-model reasoning could, in principle, be repurposed for misuse, such as the automated generation of misleading content. We do not believe these risks are differentially elevated by our contribution relative to existing distillation methods, but they remain relevant considerations for any downstream deployment.

Finally, we emphasize that the student models produced in this work are research artifacts evaluated on standard benchmarks; they have not been safety-tuned, instruction-aligned, or audited for deployment, and should not be used in user-facing systems without the additional alignment and evaluation steps that responsible deployment requires.

Taken together, we view TAD as a contribution that broadens who can meaningfully participate in language-model distillation research, while inheriting the well-documented risks of the underlying generative models it compresses.

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

## A. Derivation of the Gradient

Here we present an elaborated derivation of the gradients. The derivations follow the material in the appendix of An-shumann et al. (2025). If $p_i = \exp(z_i)/\sum_{i=1}^{|\mathcal{V}|} \exp(z_i)$ is the softmax probability for a logit $z_i$ for a vocabulary $\mathcal{V}$, then the gradient of $p_k$ is (from Iwana et al. (2019)):

$$\frac{\partial p_j}{\partial z_i} = p_j \left( \mathbb{1}_{[i=j]} - p_i \right) \tag{7}$$

Now, given a vocabulary $\mathcal{V}$, the KL Divergence loss between the teacher probabilities of the teacher $(p_i^T)$ and the student $(p_i^S)$ is:

$$\mathcal{L}_{KLD} = \sum_{i=1}^{|\mathcal{V}|} p_i^T \log(p_i^T/p_i^S) \tag{8}$$

It can be derived that,

$$
\begin{aligned}
\frac{\partial \mathcal{L}_{KLD}}{\partial z_i} &= -\sum_{j=1}^{|\mathcal{V}|} \frac{p_j^T}{p_j^S} \frac{\partial p_j^S}{\partial z_i} \\
&= -\sum_{j=1}^{|\mathcal{V}|} p_j^T \left( \mathbb{1}_{[i=j]} - p_i^S \right) \\
&= p_i^S \cdot \left( \sum_{j=1}^{|\mathcal{V}|} p_j^T \right) - \sum_{j=1}^{|\mathcal{V}|} p_j^T \mathbb{1}_{[i=j]} \\
&= p_i^S - p_i^T \tag{9}
\end{aligned}
$$

Now, we can show that $\mathcal{D}_{KL_1}$ has $K+1$ terms when we consider top-$K$ probabilities, with the first $K$ being ($i \in [K]$)

$$L_{1:K} = \sum_{k=1}^{K} \overset{*}{p}_k^T \log \frac{\overset{*}{p}_k^T}{\overset{*}{p}_k^S}$$

where $\overset{*}{p}_k^S$ are the student probabilities corresponding to the top-$K$ tokens, i.e. tokens for which the teacher probabilities are maximum. The derivative of $L_{1:K}$ w.r.t. a logit $z_i$ is

$$\frac{\partial L_{1:K}}{\partial z_i} = p_i^S \cdot \left( \sum_{k=1}^{K} \overset{*}{p}_k^T \right) - \sum_{k=1}^{K} \overset{*}{p}_k^T \mathbb{1}_{[i=k]} \tag{10}$$

Now for $i \in [\mathcal{V} \setminus K]$, the indicator function $\mathbb{1}_{[i=k]}$ is never one. Therefore, the gradient of $L_{1:K}$ has the following forms for two different cases, as:

$$
\frac{\partial L_{1:K}}{\partial z_i} = 
\begin{cases}
p_i^S \cdot (\sum_{k=1}^{K} \overset{*}{p}_i^T) - p_i^T & i \in [K] \\
p_i^S \cdot (\sum_{k=1}^{K} \overset{*}{p}_i^T) & i \in [\mathcal{V} \setminus K]
\end{cases}
$$

Note that the top $K$ probabilities do not sum to one. The last term $L_{K+1}$ can be expressed as:

$$
\begin{aligned}
L_{K+1} &= \left( 1 - \sum_{i=1}^{K} \overset{*}{p}_k^T \right) \log \frac{1 - \sum_{i=1}^{K} \overset{*}{p}_k^T}{1 - \sum_{i=1}^{K} \overset{*}{p}_i^S} \\
&= -\left( 1 - \sum_{k=1}^{K} \overset{*}{p}_k^T \right) \cdot \log \left( 1 - \sum_{k=1}^{K} \overset{*}{p}_k^S \right) + \mathrm{C}
\end{aligned}
$$

where C is a constant. The derivative of the last term, using the derivative of $p_k^S$ from Equation (7) is:

$$\frac{\partial L_{K+1}}{\partial z_i} = \frac{1 - \sum_{k=1}^{K} \mathring{p}_k^{*T}}{1 - \sum_{k=1}^{K} \mathring{p}_k^{*S}} \cdot \sum_{k=1}^{K} \frac{\partial \mathring{p}_k^S}{\partial z_i}$$

$$= \frac{1 - \sum_{k=1}^{K} \mathring{p}_k^{*T}}{1 - \sum_{k=1}^{K} \mathring{p}_k^{*S}} \cdot \sum_{k=1}^{K} \mathring{p}_k^S \left( \mathbb{1}_{[i=k]} - p_i^S \right)$$

Again, for $i \in [\mathcal{V} \setminus K]$, the indicator function $\mathbb{1}_{[i=k]}$ is never one. Therefore,

$$\frac{\partial L_{K+1}}{\partial z_i} = \begin{cases} p_i^S \cdot \left( 1 - \sum_{k=1}^{K} \mathring{p}_k^{*T} \right) & i \in [K] \\ -p_i^S \cdot \left( \frac{1 - \sum_{k=1}^{K} \mathring{p}_k^{*T}}{1 - \sum_{k=1}^{K} \mathring{p}_k^{*S}} \right) \sum_{k=1}^{K} \mathring{p}_k^{*T} & i \in [\mathcal{V} \setminus \mathcal{K}] \end{cases} \tag{11}$$

Combining the gradients of $L_{1:K}$ and $L_{K+1}$, since $\mathcal{D}_{KL_1} = L_{1:K} + L_{K+1}$

$$\frac{\partial \mathcal{D}_{KL_1}}{\partial z_i} = \begin{cases} p_i^S - p_i^T & i \in [K] \\ p_i^S \cdot \left( \frac{\sum_{k=1}^{K} \mathring{p}_k^{*T} - \sum_{k=1}^{K} \mathring{p}_k^{*S}}{1 - \sum_{k=1}^{K} \mathring{p}_k^{*S}} \right) & i \in [\mathcal{V} \setminus \mathcal{K}] \end{cases} \tag{12}$$

Therefore, the gradients of the logits corresponding to the tokens of top-$K$ teacher probabilities remain the same, while the gradients of the logits corresponding to the rest of the tokens change. The second term $\mathcal{D}_{KL_2}$ solely depends on the logits of the rest of the tokens.

$$\mathcal{D}_{KL_2} = \sum_{i \in \mathcal{V} \setminus K} \tilde{p}_i^T \log \frac{\tilde{p}_i^T}{\tilde{p}_i^S} \tag{13}$$

where we can generate $\tilde{p}_i^S$ directly from $z_i$ as $\tilde{p}_i^S = \frac{\exp z_i}{\sum_{k \in \mathcal{V} \setminus K} \exp z_k}$. Also, $\tilde{p}_i^T$ comes from a similar softmax, but is constant. Therefore,

$$\frac{\partial \mathcal{D}_{KL_2}}{\partial z_i} = \begin{cases} 0 & i \in [K] \\ \tilde{p}_i^S - \tilde{p}_i^T & i \in [\mathcal{V} \setminus \mathcal{K}] \end{cases}$$

The gradients of the logits of the top-$K$ tokens are zero for $\mathcal{D}_{KL_2}$; only their gradient for $\mathcal{D}_{KL_1}$ is non-zero (Equation (12)). And as a result, their gradient is the same as that for ordinary KL Divergence (Equation (9)). Therefore, Decoupled KD does **not** change the gradient of the logits of the top-$K$ tokens.

As for the logits of the non-top-$K$ tokens, their gradient for $\mathcal{D}_{KL_2}$ can be written as,

$$\frac{\partial \mathcal{D}_{KL_2}}{\partial z_i} = \frac{p_i^S}{1 - \sum_{k=1}^{K} \mathring{p}_k^{*S}} - \frac{p_i^T}{1 - \sum_{k=1}^{K} \mathring{p}_k^{*T}} \tag{14}$$

since $\tilde{p}_i^T = \frac{p_i^T}{1 - \sum_{k=1}^{K} \mathring{p}_k^{*T}}$ and $\tilde{p}_i^S = \frac{p_i^S}{1 - \sum_{k=1}^{K} \mathring{p}_k^{*S}}$

Therefore,

$$\left( 1 - \sum_{k=1}^{K} \mathring{p}_k^{*T} \right) \frac{\partial \mathcal{D}_{KL_2}}{\partial z_i} = p_i^S \cdot \frac{1 - \sum_{k=1}^{K} \mathring{p}_k^{*T}}{1 - \sum_{k=1}^{K} \mathring{p}_k^{*S}} - p_i^T \tag{15}$$

Combining the derivative of $\mathcal{D}_{KL_2}$ from (Equation (12)) for the tail logits, i.e., for $i \in [\mathcal{V} \setminus \mathcal{K}]$, it can easily be checked that

$$\frac{\partial \mathcal{D}_{KL_1}}{\partial z_i} + \left( 1 - \sum_{k=1}^{K} \mathring{p}_k^{*T} \right) \frac{\partial \mathcal{D}_{KL_2}}{\partial z_i}$$

$$= \left( \frac{p_i^S \cdot \sum_{k=1}^{K} \mathring{p}_k^{*T} - p_i^S \cdot \sum_{k=1}^{K} \mathring{p}_k^{*S}}{1 - \sum_{k=1}^{K} \mathring{p}_k^{*S}} \right)$$

$$+ \left( \frac{p_i^S - p_i^S \cdot \sum_{k=1}^{K} \mathring{p}_k^{*T}}{1 - \sum_{k=1}^{K} \mathring{p}_k^{*S}} \right) - p_i^T$$

$$= p_i^S - p_i^T$$

Since $\mathcal{L}_{KLD} = \mathcal{D}_{KL_1} + \left( 1 - \sum_{k=1}^{K} \mathring{p}_k^{*T} \right) \mathcal{D}_{KL_2}$, their gradients are the same. Now, for Decoupled KD, the divergence is: $\mathcal{L}_{DIV} = \mathcal{D}_{KL_1} + \beta(X) \left( 1 - \sum_{k=1}^{K} \mathring{p}_k^{*T} \right) \mathcal{D}_{KL_2}$, where $\beta(X) = \beta / (\frac{1}{N} \sum_{t=1}^{N} (1 - \sum_{k=1}^{K} \mathring{p}_k^{*T}(t)))$, where $t$ is the index of a token in a sequence $X$ containing a total of $N$ tokens. This also means,

$$\mathcal{L}_{DIV} = \mathcal{D}_{KL_1} + \left( 1 - \sum_{k=1}^{K} \mathring{p}_k^{*T} \right) \mathcal{D}_{KL_2}$$

$$(\beta(X) - 1) \left( 1 - \sum_{k=1}^{K} \mathring{p}_k^{*T} \right) \mathcal{D}_{KL_2}$$

$$= \mathcal{L}_{KLD} + (\beta(X) - 1) \left( 1 - \sum_{k=1}^{K} \mathring{p}_k^{*T} \right) \mathcal{D}_{KL_2}$$

Using Equation (15), the gradient of $\mathcal{L}_{DIV}$ has the following form for the logits $z_i$ for the tail tokens ($i \in [\mathcal{V} \setminus K]$)

$$\frac{\partial \mathcal{L}_{DIV}}{\partial z_i}$$

$$= \frac{\partial \mathcal{L}_{KLD}}{\partial z_i} + (\beta(X) - 1) \left( 1 - \sum_{k=1}^{K} \mathring{p}_k^{*T} \right) \frac{\partial \mathcal{D}_{KL_2}}{\partial z_i}$$

$$= p_i^S - p_i^T$$

$$+ (\beta(X) - 1) \left( p_i^S \cdot \frac{1 - \sum_{k=1}^{K} \mathring{p}_k^{*T}}{1 - \sum_{k=1}^{K} \mathring{p}_k^{*S}} - p_i^T \right)$$

For the logits of the top-$K$ tokens, $\frac{\partial \mathcal{D}_{KL_2}}{\partial z_i} = 0$, and therefore, their gradients are the same as those of Vanilla KD. This completes the derivation of the gradient of $\mathcal{L}_{DIV}$.

| Teacher | #P(M) | $|\mathcal{V}|$ | $d_S$ | $L_S$ | $n_H$ | $d_H$ | $d_{FFN}$ |
|---|---|---|---|---|---|---|---|
| Qwen1.5-1.8B | 1.2B | 151,936 | 1,536 | 24 | 16 | 96 | 4,224 |
| Qwen1.5-1.8B | 0.5B | 151,936 | 1,024 | 24 | 16 | 64 | 2,816 |
| Phi2-2.8B | 1.1B | 52,000 | 2,560 | 16 | 32 | 80 | 5,120 |
| Qwen2.5-3B | 1.2B | 151,936 | 2,048 | 18 | 16 | 128 | 7,680 |
| Gemma2-9B | 2B | 256,000 | 3,584 | 14 | 16 | 224 | 4,096 |

*Table 8.* The architectures of different students used in distillation for pretraining from scratch

## B. Experimental Detail

The architectures of different students for the pretraining from scratch are listed in Table 8. All students have approximately 1B active parameters, except for the 0.5B student of Qwen, which has approximately 475M active parameters. The architectures of the students of Qwen1.5 − 1.8B are kept the same as in the MiniPLM paper (Gu et al., 2025).

The experiments are divided into two major parts: pretraining distillation from scratch, and continued pretraining. For pretraining distillation from scratch, we distilled the Qwen1.5, Phi2, and Qwen2.5 models on a single H100 GPU for a week, whereas we used 2 H100 GPUs for distilling the Gemma2-9B model. We used flash attention (Dao et al., 2022) whenever possible to speed up the computation, except for Gemma2. We used Adam optimizer (Kingma & Ba, 2015) with a learning rate of $\eta = 1e - 4$ and a weight decay of $\lambda_d = 0.1$ for all the experiments. We used a batch size of 128 for all the experiments.

For the continued pretraining distillation of Tiny-Llama, we used the Adam optimizer (Kingma & Ba, 2015) with a learning rate of $\eta = 3e - 5$ and a weight decay of $\lambda_d = 0.1$ for all experiments. All experiments used a batch size of 128 and were conducted on a single NVIDIA H100 GPU. Supervised distillation is performed with a batch size of 32, $\eta = 1e - 5$, $\lambda_d = 0.1$, and a context size of 2048.

### B.1. Cost of Supervised Distillation

We conduct a comparative cost analysis of GPU hours required to produce state-of-the-art mathematical reasoning, starting with foundational models such as TinyLlama-1.1B and Llama2-7B. Models like Llemma or Rho-1 are trained with substantially higher compute budgets. Rho-1 is trained for approximately 10 hours on a 32-GPU H100 stack, requiring a total of 320 GPU hours. The best model built on Llama-7B is Llemma, which was trained on A100 GPUs for 23,000 GPU hours. Even though it uses different hardware, we can establish an equivalence by noting that the 7B model in Lin et al. (2024) takes the same number of GPU hours to train on an H100. It required 18 hours to train on 15 billion tokens using 32 H100 GPUs. Using their configuration setting, Llemma-7B will take 7,680 GPU hours to train on

a single H100. This provides a reasonable estimate, since A100s are approximately a third slower than H100 GPUs for training ($23K \approx 3 \times 7,680$). Our two-stage method requires approximately 130 hours on a single H100 GPU for TinyLlama and 420 hours on two H100 GPUs (totaling 840 hours) for Llama-2, which is substantially cheaper than the existing methods.

