# OpenReview forum: "Don't Ignore the Tail: Decoupled Distillation Produces Top Maths Students on an Academic Budget"
_ICML.cc/2026/Conference — ICML 2026 regular_

### Official Review · Reviewer_sveF · 2026-02-25

**Soundness:** 3
**Presentation:** 3
**Significance:** 3
**Originality:** 3
**Overall Recommendation:** 4
**Confidence:** 5

**Summary:**

This paper proposes Tail-Aware Distillation (TAD), a modified KL divergence objective for language model distillation that decouples the contribution of the teacher's top-K predicted probabilities from the remaining tail distribution. The key observation is that standard KL divergence is dominated by the teacher's highest-probability modes, suppressing the learning signal from the tail of the distribution. TAD amplifies the tail's contribution by multiplying the tail KL term with a hyperparameter $\beta$, normalized by the sequence-level mean of the tail probability mass for stability. The method is computationally efficient (similar FLOPs to vanilla KD) and is demonstrated across pretraining distillation (Qwen1.5-1.8B, Phi-2, Qwen2.5-3B, Gemma2-9B to ~1B students), continued pretraining (Phi3-4B → TinyLlama), and supervised distillation for mathematical reasoning, all within a constrained academic compute budget.

**Compliance With Llm Reviewing Policy:**

Affirmed.

**Key Questions For Authors:**

1. How does TAD perform in longer training regimes (e.g., 25B or 50B tokens)? The current experiments use only 2B tokens, and it is possible that the vanilla KD student catches up with TAD given more training. Understanding the scaling behavior of TAD is important for assessing its long-term utility.

2. Have the authors considered combining TAD with on-policy distillation methods? Since TAD operates at the divergence level and on-policy methods operate at the sampling level, they seem conceptually orthogonal. Preliminary results on such a combination could significantly increase the impact of the paper.

3. The paper assumes $\beta = 2$ as the default. How does performance degrade for temperatures other than 1? If the teacher's distribution is already softened via temperature scaling, the tail probability mass increases naturally. Does TAD still provide benefits when combined with temperature-based distillation?

**Limitations:**

The authors do not explicitly discuss limitations. Key limitations that should be acknowledged include: the limited training budget (2B tokens), the focus on offline distillation only, the lack of comparison with on-policy methods, and the unclear scaling behavior of TAD with larger compute budgets.

**Strengths And Weaknesses:**

### Strengths

1. The method is simple, well-motivated, and computationally efficient. The decoupling of top-K and tail contributions to KL divergence is a clean idea with solid gradient analysis (Section 2.1) showing how the tail gradients are amplified. The fact that TAD has essentially the same FLOP count as vanilla KD (Table 3) while consistently outperforming it makes it practical for adoption.

2. The experimental breadth is commendable. The paper distills models from multiple families (Qwen, Phi, Gemma) across multiple settings (pretraining, continued pretraining, supervised distillation), demonstrating that TAD generalizes across diverse teacher-student configurations. The comparison with MiniPLM is particularly informative, showing that TAD achieves better results with significantly less compute.

3. The paper includes useful practical analyses: calibration error evaluation (Full-ECE), FLOP comparisons, and the selection of $K$ analysis linking performance to the Zipfian nature of next-token distributions. The connection to DKD [1] and the clear articulation of why DKD's label-anchored decoupling is not suitable for LM distillation (due to the 39%–46% mismatch rate between ground-truth tokens and teacher modes) is well-argued.

### Weaknesses

1. The paper focuses entirely on offline distillation, whereas the LLM distillation landscape has been moving towards on-policy methods (e.g., GKD [2], DistiLLM [3]) which have demonstrated stronger performance by addressing distribution mismatch between teacher and student generation. The paper should discuss how TAD relates to or could be combined with on-policy approaches, and include comparisons with at least one on-policy distillation baseline.

2. The experiments use only 2B tokens for pretraining distillation, which is very limited by modern standards. While the paper frames this as a compute-efficient approach for academic budgets, the evaluation models are consequently weak in absolute terms. It is unclear whether the relative improvements of TAD over vanilla KD would persist or diminish with larger training budgets (e.g., 50B+ tokens), where the student has more opportunity to learn the full distribution.

3. The gradient analysis (Section 2.1) assumes the student's top-K probability mass approaches 1 ($\sum_{k=1}^{K} p_k^S \approx 1$) to justify the amplification mechanism. However, this condition may not hold throughout training, particularly in early stages when the student is poorly calibrated. A more thorough analysis of when the tail amplification is beneficial versus harmful across different training stages would strengthen the theoretical contribution.

4. The paper mentions DistilLM (Ko et al., 2024) briefly, noting it "produces results similar to MiniLLM while reducing execution time," but does not provide a direct comparison. Given that DistilLM addresses a similar problem domain (efficient distillation) and that the reviewer is familiar with this work, a direct experimental comparison would be more informative.

[1] Zhao et al., "Decoupled Knowledge Distillation." CVPR. 2022
[2] Agarwal et al., "On-Policy Distillation of Language Models: Learning from Self-Generated Mistakes." ICLR. 2024
[3] Ko et al., "DistiLLM: Towards Streamlined Distillation for Large Language Models." ICML. 2024

---

> ### Author Rebuttal · Authors · 2026-03-30
>
> We thank Reviewer sveF for their thorough assessment. We hope our response clarifies the questions as mentioned below.
>
>
> W1:
> We agree that on-policy methods (e.g., GKD, DistiLLM) are an important direction. But they are computationally expensive due to repeated sampling from the student, and require 3.5-4.2 $\times$ the number of FLOPs compared to Vanilla KD for the distillation of Qwen1.5 (Table 3). They are highly unsuitable for large scale pretraining distillation.
> Our focus in this work is on offline distillation, and we focus on eliminating the effect of the strong teacher modes in regular KL divergence with a tail-aware divergence, which has a similar computational profile. TAD can also be directly combined with on-policy approaches without additional computational overhead in the loss computation
>
>
> W2:
> We agree that understanding scaling behavior is important. Our current setup intentionally focuses on compute-constrained regimes to reflect realistic academic settings. We can use the Chinchilla scaling law (https://arxiv.org/abs/2203.15556) to estimate the scaling capability of the methods up to 1T tokens, using the equation $L(N) = C + \frac{A}{N^\alpha}$, where N is the number of training tokens, $L(N)$ is the corresponding loss, and $C$ and $A$ are constants. When we compute the loss on the validation set of the Regmix dataset, we can see that TAD consistently achieves the lowest loss in the experimental setup of Table 4.
>
> __Phi2.8B $\rightarrow$ 1B__
>
> |No of Tokens| 10B |100B | 1T|
> |:---:|:---:|:---:|:---:|
> |Vanilla KD| 2.80 | 2.77 | 2.76 |
> |RKL | 2.91 | 2.86 | 2.84
> |TAD(K=10) | 2.78 | 2.73 | 2.71
>
> __Qwen2.5 3B $\rightarrow$ 1B__
>
> |No of Tokens| 10B |100B | 1T|
> |:---:|:---:|:---:|:---:|
> |Vanilla KD| 3.18 | 3.09| 3.05|
> |RKL | 3.26 | 3.15 | 3.11 |
> |TAD(K=10) | 3.04 | 2.92 | 2.87|
>
> __Gemma 9B $\rightarrow$ 2B__
>
> |No of Tokens| 10B |100B | 1T|
> |:---:|:---:|:---:|:---:|
> |Vanilla KD| 3.15 | 3.00| 2.93|
> |RKL | 3.23 | 3.08 | 3.01 |
> |TAD(K=10) | 3.08 | 2.94 | 2.88|
>
>
> W3:
> In most of the experiments, the student is typically initialized from the teacher's weights (truncated/copied, as in Section 3.2), but with randomly initialized MLP layers. As a result, the student's token probabilities are poorly calibrated. In this regime, $p^S_k$ is initially much flatter than $p^T_k$ for most tokens, and the gradient$\frac{\partial{L}_KLD}{\partial z_i} = p^S_i − p^T_i \approx - p^T_i  $. Since $p^T_k >> p^T_i$ for top-K tokens ($k \in [K]$) vs. tail tokens ( $i \notin [K]$), the gradient magnitudes for top-K tokens vastly exceed those for tail tokens during early training when most learning occurs. This drives the student's top-K probability mass rapidly toward 1.
> TAD’s compensation mechanism activates when the student overestimates top-K mass relative to the teacher, and naturally diminishes as the distributions align. Thus, the method adapts across training stages rather than relying on a fixed assumption.
>
> W4:
>
> Although DistilLM tries to make MiniLLM efficient, it is still an on-policy algorithm requiring samples to be drawn from the student model. Figure 2.5(d) of DistilLM (https://arxiv.org/pdf/2402.03898) shows that it is 2-2.5x slower than Vanilla KD, while MiniLLM is 7x slower. Here, our focus is to come up with a better divergence with a similar computational profile as the Vanilla KD. We focus on offline distillation for computational efficiency; comparison with on-policy methods is an important direction for future work.
>
> Q1: Please refer to W2
> Q2: It is possible to combine TAD divergence with on-policy distillation. However, the main focus of our paper is pretraining distillation, for which the on-policy distillation can be very computationally expensive (refer to the response to W1 in detail). This is why we keep such experiments out of scope of this paper.
>
> Q3:
> We included a study on the sensitivity of $\beta$ for the distillation of Qwen1.5, and included the results in Table 2. The performance peaks at 2, with a smooth degradation on either side. We kept $\beta=2$ for the rest of the experiments.
>
> When coming to temperature, we fix $\tau=1$ to isolate the effect of the divergence. Temperature scaling increases tail mass by globally smoothing the distribution, whereas TAD selectively reweights the tail relative to the top-K region.
> Thus, the two mechanisms are complementary: while temperature controls global entropy, TAD controls relative weighting between modes and tail. Temperature scaling and TAD operate on different axes:
> 1.	Temperature globally smooths the distribution, flattening both top-K and tail probabilities uniformly via the softmax.
> 2.	TAD selectively reweights the tail's contribution to the loss while leaving the top-K gradient unchanged (Section 2.1).
> We expect TAD to work even when we use a temperature $\tau>1$. Even if the teacher probabilities are smoothened through temperature, the relative reweighting of the top and the tail divergences will still be beneficial.

---

> > ### Author Rebuttal · Reviewer_sveF · 2026-04-03
> >
> > The authors have adequately addressed my main concerns. For W1 and W4, the computational argument is compelling — on-policy methods require 2–4× more FLOPs than vanilla KD, making them genuinely impractical in the large-scale pretraining regime this paper targets. The clarification that DistiLLM is also on-policy further justifies its exclusion from direct comparison. For W2/Q1, the Chinchilla scaling extrapolation is reasonable and shows TAD's advantage over vanilla KD persisting consistently at 10B, 100B, and 1T tokens across all three teacher-student pairs, which sufficiently addresses my concern about whether the gap closes with more training. For W3, the authors' explanation of early training dynamics (student initialized from teacher with randomly re-initialized MLP layers, yielding an initially flat distribution) clarifies that the top-K probability mass converges toward 1 rapidly in early training, making the asymptotic assumption effectively self-fulfilling rather than a hard constraint. For Q3, the argument that temperature scaling and TAD operate on orthogonal axes — global entropy smoothing vs. selective top-K/tail reweighting — is conceptually sound.
> >
> > I maintain my score of 4 (weak accept). The core limitations (2B token training budget, no empirical on-policy combination) remain, but they are now clearly framed and justified within the paper's stated scope of compute-constrained academic distillation.

---

### Official Review · Reviewer_VB4s · 2026-03-09

**Soundness:** 3
**Presentation:** 2
**Significance:** 3
**Originality:** 3
**Overall Recommendation:** 5
**Confidence:** 4

**Summary:**

This paper proposes Tail-Aware Distillation (TAD), a modification of the standard token-level KL objective for language model distillation that explicitly decouples the teacher’s top-K probability mass from the tail and amplifies the contribution of the tail via a normalized weighting. TAD reduces the impact of the teacher modes and, consequently, increases the contribution of the tail of the distribution. Experimental results demonstrate that the modified distillation method(TAD) yields competitive performance in both pre-training and supervised distillation of decoder models across various datasets.

**Compliance With Llm Reviewing Policy:**

Affirmed.

**Final Justification:**

The rebuttal addresses most of my concerns.

**Key Questions For Authors:**

1. How sensitive is TAD to the teacher’s temperature during distillation?
2. Can you add ablation study of only top-K token prob?

**Limitations:**

1. The improvement of performance is fair.

**Strengths And Weaknesses:**

Strengths
1. Evaluations span multiple teachers and includes compute analysis (PFLOPs/GPU-hours) showing TAD’s cost is close to vanilla KD, which is solid.
 2. The proposed method is simple but works, can be generalized to other models.
TAD maintains the integrity of the teacher’s whole distribution while strategically rebalancing the objective. By utilizing a top-K vs. tail split, TAD prevents dominant high-probability tokens from overwhelming the gradient signal. Furthermore, the inclusion of sequence-level normalization ensures training stability, shifting the student's focus toward the informative "tail mass" without compromising the global distribution's numerical consistency.




Weaknesses
1. Lack of the Temperature variation. In standard Knowledge Distillation, the temperature(T) is used to smooth the output token probability distribution, T can inflate the tail, so investigating the interaction between T and rank-based decoupling is essential

2. Lack of impact statement.
3. Lack of ablation study of only top-K token prob without the tail prob to further justify your method.  To justify your method, you should add the top-K only ablation as a "loss of dark knowledge" experiment. This demonstrates that ignoring the tail isn't just a minor omission, but it's a fundamental degradation of the teacher's guidance. And you need to add more explanation.

---

> ### Author Rebuttal · Authors · 2026-03-30
>
> We thank Reviewer VB4s for their thorough assessment. We hope our response clarifies the questions as mentioned below.
>
> W1:
> We agree that temperature is an important factor in distillation. In our experiments, we fix the temperature $\tau=1$ to isolate the effect of the divergence, this is also the default in all baselines we compare against (Vanilla KD, MiniPLM, RKL). In fact, most of the existing work on LM distillation keeps $\tau=1$ . Temperature scaling and TAD operate on different axes:
> 1.	Temperature globally smooths the distribution, flattening both top-K and tail probabilities uniformly via the softmax.
> 2.	TAD selectively reweights the tail's contribution to the loss while leaving the top-K gradient unchanged (Section 2.1).
> Crucially, higher temperature increases tail mass but also flattens the top-K structure, losing the discriminative signal among the teacher's most confident predictions, namely, the modes. TAD avoids this trade-off by amplifying the tail without distorting the modes. The two mechanisms are therefore different, and we expect TAD to remain beneficial even with $\tau >1$. We will include temperature ablations and a discussion of this interaction in the camera-ready version.
>
> W2:
>
> We appreciate this suggestion and will include a broader discussion of impact in the camera-ready version. In particular, TAD lowers the computational barrier for effective distillation, enabling smaller models to benefit from large teachers under modest academic budgets. This has implications for accessibility, energy efficiency, and deployment on resource-constrained devices.
>
> W3:
> Conceptually, removing the tail corresponds to discarding $D_{KL_2}$ entirely, retaining only $D_{KL_1}$. This not only eliminates the corrective tail gradient signal but also changes the fixed point of the algorithm. From Eq. (12), the gradient of $D_{KL_1}$ for tail logits ($i \in \mathcal{V} \setminus K$) is:
>
> $$\frac{\partial D_{KL_1}}{\partial z_i} =
> p^S_i \cdot \frac{\sum_{k=1}^{K} p_k^{\ast T} -
> \sum_{k=1}^{K} p_k^{\ast S}}{1 - \sum_{k=1}^{K} p_k^{\ast S}}$$
>
> This gradient vanishes when the aggregate top-K masses match ($\sum_{k=1}^{K} p_k^{\ast T} = \sum_{k=1}^{K} p_k^{\ast S}$), regardless of whether individual tail probabilities match the teacher. The fixed point is therefore __not__ $p^S_i = p^T_i$ for tail tokens, only the aggregate tail mass is constrained, leaving individual tail token probabilities unconstrained. This constitutes incorrect convergence for the tail distribution, which will result in the miscalibration of the students. This phenomenon is empirically observed in Anshumann et al. (ACL 2025: https://arxiv.org/abs/2503.16870), who find that top-K-only distillation significantly reduces calibration and distributional fidelity, which is fully consistent with this analysis.
>
>
> __Limitation__: "The improvement of performance is fair."
>
>
> Since we are using pretraining distillation from scratch using only 2-2.5B tokens, the improvements are modest. Our aim here is to benchmark the method against existing methods, where we show consistent performance gain across models, datasets, and settings (Tables 1, 4, and 5), including more challenging benchmarks. However, pretraining, with or without distillation, cannot achieve a significant performance gain without further fine-tuning.
>
> Furthermore, in supervised distillation settings, the gains are substantial. For mathematical inference (Section 3.4), TAD improves the GSM8K score of TinyLlama from 2.0 to 36.8 (Table 6), while the average score for math tasks increases from 11.4 to more than 42. Further, a similar experiment on Llama2-7B elevates the GSM8K score from 14.2 to 56.0, with the average score increasing from 31.4 to 54.0. These results show that our method achieves substantial performance improvements when used for supervised distillation
>
>
>
> Q1: Please refer to the answer of W1.
>
> Q2: Please refer to the answer of W3.

---

> > ### Author Rebuttal · Reviewer_VB4s · 2026-04-03
> >
> > The review addresses most of my concerns. I believe this work clears the bar for acceptance, especially if the results presented during the rebuttal are incorporated into the paper.

---

> > > ### Author Response · Authors · 2026-04-03
> > >
> > > Thanks for the response. We are pleased that we were able to address most of your concerns. The results of Llama2-7B for supervised distillation in mathematical reasoning are available in Table 8 of Appendix B. We will include the rebuttal results in the camera-ready version.
> > >
> > > Meanwhile, if you feel that the work clears the acceptance bar, we would greatly appreciate it if you could consider updating your score to reflect your current assessment. Thank you again for your valuable feedback, which has helped strengthen our paper.

---

### Official Review · Reviewer_WDuf · 2026-03-11

**Soundness:** 3
**Presentation:** 4
**Significance:** 4
**Originality:** 3
**Overall Recommendation:** 5
**Confidence:** 4

**Summary:**

This submission proposes Tail-Aware Distillation (TAD), a novel objective function for distilling causal language models. The authors identify that standard KL divergence is often dominated by a teacher's highest-probability modes, leading to the neglect of informative lower-probability tokens in the "tail" of the distribution. TAD decouples the teacher's top-K predictions from the rest of the vocabulary, applying a probability-mass-normalized tail KL divergence that amplifies the contribution of tail tokens. The method is optimized for "modest academic budgets," allowing distillation of billions of tokens within a week on a single GPU. Experiments on Qwen, Phi, and Gemma models demonstrate that TAD consistently outperforms vanilla KD and data-centric methods like MiniPLM across pre-training and supervised mathematical reasoning tasks.

**Compliance With Llm Reviewing Policy:**

Affirmed.

**Key Questions For Authors:**

1. Table 4 shows that "F-ECE" (calibration error) increases slightly as $K$ increases from 1 to 20. Can the authors explain the theoretical mechanism behind this increase in calibration error?
2. In the gradient analysis, the sequence-specific $\beta(X)$ is introduced. Does the variance of this term across different batches affect training stability for very large models?
3. Why does the performance of TAD peak at $K=10$ and then decline at $K=20$? Is this related to the teacher's entropy or the noise levels in the tail?

**Limitations:**

The authors mention the inherent limitation of distillation compared to training from scratch and the challenge of distilling from closed-source corpora. They could further discuss how TAD performs on non-English corpora where the tail distribution might differ significantly.

**Strengths And Weaknesses:**

Soundness:
The technical claims are well-supported by a combination of gradient analysis and extensive empirical results. The authors provide a compelling derivation in Section 2.1 showing how TAD creates a "gradient compensation" effect that drives up the student's tail probability mass as long as it deviates from the teacher's distribution. The observation that teacher modes frequently mismatch the ground-truth next token (39% to 46% mismatch rate) justifies the focus on distributional learning over label-anchored decoupling. The experiments are well-designed, including sensitivity analysis for the hyperparameters $K$ and $\beta$.
Presentation:
The paper is clearly written and well-structured. Figure 1 effectively visualizes the convergence speed of TAD compared to vanilla KD, and Figure 2 provides intuitive evidence of the "Zipfian" nature of token distributions that TAD aims to exploit. The comparison with SOTA models like Rho-1-Math and Llama3.2-1.2B is clear and contextually grounded.
Significance:
This manuscript addresses a crucial practical problem: making high-quality distillation accessible to researchers without industry-scale compute resources. The "academic budget" focus is highly relevant and likely to influence future research in small language models (SLMs). The demonstration that TAD can significantly improve mathematical reasoning in models as small as TinyLlama-1.1B is a notable contribution.
Originality:
While inspired by Decoupled Knowledge Distillation (DKD), TAD's adaptation to causal, label-free scenarios using rank-anchored (top-K vs. tail) partitioning is a significant original contribution. The introduction of sequence-level normalization to stabilize the tail loss is a creative solution to the convergence issues encountered with simpler decoupling formulations.

---

> ### Author Rebuttal · Authors · 2026-03-30
>
> We thank Reviewer WDuf for the thorough and positive assessment. We are glad the contributions on soundness, significance, and originality were well-received. We address the three key questions below.
>
> Q1:
>
> As $K$ increases, the tail mass $\alpha_K^T = 1 - \sum_{k=1}^{K} p_k^T$ decreases sharply due to the Zipfian nature of token distributions (Figure 2). For many tokens, particularly low-entropy ones, $\alpha_K^T(t) \rightarrow 0$ at large $K$. This shrinks the sequence-level mean $\bar{\alpha}_K^T$ in the denominator of $\beta(X) = \beta / \bar{\alpha}_K^T$, causing $\beta(X)$ to grow large. The amplified tail KL term is then dominated by the small number of remaining high-entropy tokens, introducing slight over-dispersion in the predicted probabilities, which manifests as a modest F-ECE increase. Importantly, even at $K = 20$, TAD remains better calibrated than all baselines including Vanilla KD and RKL (Table 4), confirming the effect is limited and controlled.
>
>
> Q2:
> The sequence-specific scaling factor $\beta(X)$ introduces mild variance across batches; however, two design choices ensure stability:
>
> 1. __Normalization by mean tail mass:__ $\beta(X) = \beta / \left(\frac{1}{N}\sum_t \alpha_K^T(t)\right)$ reduces sensitivity to per-token fluctuations. If a batch has, say, 32 to 128 samples, each with around 1024 tokens, this means it is being averaged across 32000 to 128000 tokens. This reduces the local variation in $\beta(X)$ significantly.
>
> 2. __Bounded operating regime:__ Since $\alpha_K^T(t) \in (0,1)$, $\beta(X)$ remains well-behaved in practice.
>
> Empirically, we observe stable training across all experiments without additional tuning. We will clarify this point and include a brief variance analysis in the camera-ready version.
>
> Q3:
> The explanation lies in the noise structure of the tail, not teacher entropy per se. As shown in Figure 2(a), $\alpha_K^T$ declines sharply beyond $K \approx 5$--$10$: most probability mass is concentrated in the top tokens, leaving the tail dominated by near-uniform, high-entropy tokens for larger $K$. The normalised tail KL term $D_{KL2}$ therefore increasingly amplifies noise rather than informative secondary predictions, yielding diminishing returns. Crucially, the optimal range $K = 5$--$10$ is consistent across all four teacher families: Qwen1.5, Phi-2, Qwen2.5, and Gemma2, despite their substantially different sizes and entropy profiles (Tables 1 and 4). If teacher-specific entropy were driving the plateau, we would expect different optimal $K$ values per teacher. The cross-model consistency instead confirms it is the shared Zipfian noise structure of the tail that sets the effective limit.

---

### Official Review · Reviewer_eeQp · 2026-03-12

**Soundness:** 2
**Presentation:** 2
**Significance:** 2
**Originality:** 2
**Overall Recommendation:** 3
**Confidence:** 3

**Summary:**

This manuscript proposes Tail-Aware Distillation (TAD), a new knowledge distillation objective for language models. The authors identify a limitation in KL divergence where the loss is dominated by the teacher’s high probabilities, often causing the student to ignore the tail of the distribution. TAD decouples the teacher's top-K probabilities from the rest of the distribution, amplifying the learning signal from the tail without increasing the computational profile of standard KL. This manuscript includes several experimental results.

**Compliance With Llm Reviewing Policy:**

Affirmed.

**Final Justification:**

This paper proposes a new knowledge distillation objective function for language models, TAD. The authors point out a limitation of forward KL divergence, where it is dominated by the teacher’s high-probabilities, causing the student model to ignore the tail of the distribution. TAD separates the teacher’s top-K probabilities from the rest of the distribution, thereby enhancing the learning signal in the tail region without increasing the computational complexity of forward KL.

In my review, I raised questions about (1) mismatched introduction, (2) evidence for tail distribution matching, (3) comparison with other divergences, (4) Lack of novelty, and (5) further clarifications. Most concerns, with the exception of (3), have been addressed through the rebuttal. The authors provided additional responses for (3), but these still did not include practical evidence. Therefore, while I have raised the initial score, I have determined that it is difficult to provide a higher score.

**Key Questions For Authors:**

1. I am curious about the derivation process of Eq. (2), which splits the KL divergence into the top-K portion and the remaining portion.
2. In Eq. (5), the gradient of KL is expressed as the difference between the student probability and the teacher probability. However, I'm curious why it's claimed that top-K probability tokens have larger gradients than tail tokens. Even if the teacher probability has a large value, if the student probability is large, wouldn't the gradient size be small?

**Limitations:**

Please see the weaknesses.

**Strengths And Weaknesses:**

**Strengths**

1. Proposed method is simple and practically appealing. It is easy to understand, sits very close to standard KD loss function, which enables easy substitution.
2. Required additional cost is small, which is great advantage in practice. The manuscript reports that TAD has nearly the same PFLOP cost as vanilla KD, and makes a similar in the larger-model setting.

**Weaknesses**

1. The core claim of this manuscript and its introduction are mismatched. While the manuscript's title and abstract emphasize the need to address the tail of the distribution, the introduction mentions the computational challenges of LLM distillation. I believe there is a logical gap between these two claims.
2. There is insufficient concrete evidence that improving matching in the tail of the distribution corresponding to low probability helps enhance LLM distillation performance. Theoretically, any divergence also guarantees matching of tail probabilities. If so, the proposed method might be satisfy tail probability matching at an earlier stage of training. However, there is no detailed analysis explaining why this characteristic aids LLM distillation performance or why the proposed method exhibits this feature. Furthermore, it remains unproven whether matching the tail of the distribution negatively impacts matching the top-K probabilities.
3. Despite the existence of various previous studies using divergences other than KL divergence, this manuscript only compares its results with KL divergence. Other divergences may already perform well in matching tail probabilities. However, the lack of comparison with diverse divergences in this manuscript makes it difficult to fully appreciate the effectiveness of the proposed method.
4. Lack of novelty. TAD is a clean and useful modification, but conceptually it is still a fairly local reweighting of KL: split top-K vs. tail, renormalize, and scale the tail. The manuscript is transparent that it is inspired by DKD, but the jump from DKD-style decoupling to rank-based LM distillation feels incremental rather than fundamentally new. The contribution is more of a strong engineering refinement than a major algorithmic advance.

---

> ### Author Rebuttal · Authors · 2026-03-30
>
> We thank the reviewer eeQP for the careful reading and constructive feedback.
>
> W1: The paper's scientific motivation is that standard KL divergence is dominated by the top-K probabilities of the teachers, and TAD corrects this. Efficiency is complementary: since pretraining distillation runs over billions of tokens, any useful divergence must maintain similar FLOPs as the KL divergence, which TAD achieves (Table 3). We will restructure the introduction to lead with tail-awareness and position efficiency as an enabling property.
>
>
> W2:
> Our claim concerns the convergence rate, not the fixed point. From Eq. 5, the KL gradient is $\frac{\partial{L}_{KLD}}{\partial z_i} = p^S_i − p^T_i$. Due to the skewed teacher distribution, top-K tokens dominate gradients early in training, driving $\sum p^S_k \approx 1$ and suppressing tail learning. TAD's compensation term (Eq. 6) activates when the student overestimates top-K mass, amplifying tail gradients while preserving the same fixed point. Empirically, TAD achieves lower held-out KL than vanilla KD even though KL is the evaluation metric (Fig. 1), and improves both accuracy and calibration across all four teacher families (Table 4), confirming that better tail matching yields better-calibrated models.
>
> Furthermore, as shown in Section 2.1 (line 156), TAD does not change the gradient of top-K logits — only tail logits receive additional signal. The top-K gradient remains $p^S_i − p^T_i$, identical to standard KL. Improved tail matching comes at no cost to top-K convergence.
>
> If the question is why include the tail divergence at all, we explained it in the answer of W3 of VB4s due to space constraints.
>
>
>
> W3:
> We also compare against an additional divergence measure: Reverse KL (RKL), which is reported in Table 4 alongside Vanilla KD, MiniPLM, and TAD. RKL has been widely used in recent LM distillation literature (MiniLLM, on-policy KD). As shown in Table 4, RKL yields the worst calibration among all methods, consistent with its mode-seeking behaviour. TAD outperforms RKL without affecting the calibration. We compare TAD with all the popular divergences used in the LM distillation literature.
>
> Regarding other divergences, such as JS or f-divergences more broadly, these were not included because the primary contribution of this paper is a specific, lightweight modification to the forward KL divergence.
>
>
> W4:
> The paper is transparent about DKD (Zhao et al., 2022), inspiring the decoupling idea, but we argue that TAD is a conceptually distinct contribution because it performs rank-based decoupling of KL divergence, whereas DKD is label-anchored. We submitted the paper to the “Large Language Models” category, not to generic ML; the adaptation to Language Model distillation is the main focus of our paper. At a 39–46% label-mismatch rate (Fig. 2b) between the teacher’s model and the next token, direct adaptation of DKD would produce conflicting gradients.
>
>
> Q1:
> $$D_{KL}(P^T || P^S) = \sum_{i \in V} p^T_i \log \frac{p^T_i}{p^S_i}$$
>
> Eq.(2) follows directly from partitioning the vocabulary $V$ into the top-$K$ set and its complement. The full KL divergence is:
>
> If the probability of the top-$K$ tokens are $ \\{ p_k^{*T} \\}_{k=1}^K $,
>
> $$D_{KL}(P^T || P^S)= \sum_{k=1}^{K} p_k^{*T} \log \frac{p_k^{*T}}{p_k^{*S}}+ \sum_{i \notin [K]} p^T_i \log \frac{p^T_i}{p^S_i}$$
>
> Define: $\alpha^T_K = 1 - \sum_{k=1}^{K} p^T_k$
>
> Then $\tilde{p}^T_i = \frac{p^T_i}{\alpha^T_K}$
>
> $$\sum_{i \notin [K]} p^T_i \log \frac{p^T_i}{p^S_i} = \alpha^T_K \underbrace{\sum_{i \notin [K]} \tilde{p}^T_i \log \frac{\tilde{p}^T_i }{\tilde{p}^S_i } }_{ D(KL2) } + \alpha^T_K \log \frac{ \alpha^T_K }{ \alpha^S_K }$$
>
> If we include $\alpha^T_K \log \frac{ \alpha^T_K }{ \alpha^S_K }$ in $D_{KL1}$, we can write
>
> $$D_{KL}(P^T || P^S) = D_{KL1} + \alpha^T_K D_{KL2}$$
>
>
>
>
>
> Q2: The KL gradient $\frac{\partial{L}_{KLD}}{\partial z_i} = p^S_i − p^T_i$, so the gradient for a top-K token k is large only if $|p^S_k − p^T_k|$ is large. If the student has already matched the teacher's top-K probability, the gradient would be small.
>
> The claim in Section 2.1 is about the early-to-mid training regime, before the student has converged. The student is typically initialized from the teacher's weights, but with randomly initialized MLP layers. As a result, the student's token probabilities are poorly calibrated, making $p^S_k$ initially much flatter than $p^T_k$ for most tokens, and the gradient $\frac{\partial{L}_{KLD}}{\partial z_i} \approx −p^T_k$. Since $p^T_k \gg p^T_i$ for top-K tokens $k \in [K]$ vs. tail tokens $ i \notin [K]$, gradient magnitudes for top-K tokens vastly exceed those for tail tokens during early training when most learning occurs. This drives the student's top-K probability mass rapidly toward 1, after which the tail gradients under vanilla KL remain very small (since both $p^S_i$ and $p^T_i$ are near-zero for tail tokens). We will clarify this in the final version

---

> > ### Author Rebuttal · Reviewer_eeQp · 2026-04-03
> >
> > I have read the rebuttal and appreciate the authors’ additional clarifications. Most of my concerns have been addressed, and I have updated my score accordingly.
> >
> > However, the following concerns are not resolved.
> >
> > **Comparison with RKL**
> >
> > The authors state that they compare against RKL in Table 4. However, RKL achieves relatively high average performance than several other baselines in Table 4, which raises the question of why RKL is omitted in other result tables. If average performance is considered an core evaluation criterion, then a more consistent comparison with RKL should be included throughout. Conversely, if the tables are not intended to emphasize average performance, then I believe they also report the corresponding F-ECE performance column for completeness. In my opinion, including both would substantially strengthen the empirical support for the proposed method.
> >
> > **Comparison with other divergences**
> >
> > The authors argue that comparisons beyond RKL were not included because the primary contribution of this work is a lightweight modification of forward KL divergence. However, I do not find this justification sufficient to establish the practical effectiveness of the proposed method. In particular, if an alternative divergence without TAD achieves better performance than Forward KL + TAD, then the motivation for using the proposed method becomes weaker. To address this concern and more convincingly demonstrate the value of the method, the paper should compare against a broader range of divergences used in recent LM distillation work, including RKL and other divergences adopted in previous studies [1, 2, 3].
> >
> > The authors also mention resource limitations (Section 3.2.1) and similar computational efficiency (as discussed in W4 of Reviewer sveF) as reasons for not comparing against DistiLLM. I think this explanation unconvincing for two reasons. First, the paper already includes comparisons with MiniLLM, which is substantially slower than Vanilla KD (reported as 7x slower), while DistiLLM is reported to be only 2.5x slower than Vanilla KD. Second, the paper does not compare against the skewed KL divergences used in DistiLLM, which are particularly relevant to the present work. Finally, even if resources are limited, the gradient analysis (Section 2.1) on other divergences still be feasible and would already provide useful evidence.
> >
> > **Compatibility with other divergences**
> >
> > Related to the above, the authors’ response also raises a broader question regarding the compatibility of TAD with divergences other than forward KL. Do other divergences also suffer from the teacher’s top-K probabilities? If so, it would be important to understand whether combining those divergences with TAD also leads to consistent gains. Clarifying this point would help position the method more clearly and would also strengthen the claim that the proposed approach addresses a general optimization issue rather than one specific to forward KL alone.
> >
> > [1] On-Policy Distillation of Language Models: Learning from Self-Generated Mistakes, ICLR 2024
> >
> > [2] DistiLLM: Towards Streamlined Distillation for Large Language Models, ICML 2024
> >
> > [3] ABKD: Pursuing a Proper Allocation of the Probability Mass in Knowledge Distillation via α-β-Divergence, ICML 2025

---

> > > ### Author Response · Authors · 2026-04-07
> > >
> > > We thank Reviewer eeQp for the detailed follow-up. We address the remaining concerns below.
> > >
> > > __1. Comparison with RKL__
> > >
> > > The reviewer is correct to note that we discard RKL beyond the experiments in Table 4, as it degrades calibration. As shown in Anshuman et al. (ACL 2025: https://arxiv.org/abs/2503.16870), a miscalibrated model might perform better on few-shot tasks. But when they are fine-tuned on downstream tasks or further instruction-tuned, their performance degrades. We will add the RKL results to Tables 1 and 6 in the camera-ready version.
> > >
> > > __2. Comparison with other divergences__
> > >
> > > The primary hypothesis of the paper is that the KL divergence is heavily dominated by the teacher's modes, and that addressing the mode dominance in the teacher's forward KL divergence by decoupling the head and tail divergences yields subsequent improvement. We demonstrated that our results are competitive with or better than those of the most commonly used divergence in LM distillation. Our contribution is a specific, principled correction to forward KL's mode dominance, not a claim of optimality across all divergence families.
> > >
> > > On-policy distillation algorithms like GKD, MiniLLM, and DistilLLM are a different group of algorithms and are primarily used in supervised settings. They are not straightforwardly comparable to our work, and it is quite possible to design an on-policy distillation using TAD rather than the KL divergence. As for MiniLLM experiments, we present the FLOP counts in Table 3. It shows how costly MiniLLM can be for pretraining distillation on Qwen1.8B. For the larger teachers, it is not feasible to run MiniLLM under the settings in Table 4 using our resources.
> > >
> > >
> > > __3. Compatibility with other divergences__
> > >
> > > This is an insightful question. Other divergences might suffer from top-K dominance. For example, the α-β divergence (e.g., as used in ABKD) offers a smooth interpolation between forward and reverse KL, which turns into forward KL for $\alpha \rightarrow 1, \beta \rightarrow 0$ and reverse KL for $\alpha \rightarrow 0, \beta \rightarrow 1$. If we look into it,
> > >
> > > $$ D^{\alpha,\beta}(P^T || P^S) = -\frac{1}{\alpha\beta}  \sum_j  \Big[
> > > (p_j^T)^{\alpha} (p_j^S)^{\beta} - \frac{\alpha}{\alpha+\beta} (p_j^T)^{\alpha+\beta} - \frac{\beta}{\alpha+\beta} (p_j^S)^{\alpha+\beta} \Big]
> > > $$
> > > Thus $$ D^{\alpha,\beta}(P^T || P^S)  = D_1 (P^T , P^S) + D_2 (P^T) + D_3 (P^S) $$
> > >
> > > where
> > > $$
> > > D_1(P^T, P^S) = -\frac{1}{\alpha\beta} \sum_j (p_j^T)^\alpha (p_j^S)^\beta,
> > > $$
> > > $$
> > > D_2(P^T) = \frac{1}{\beta(\alpha+\beta)} \sum_j (p_j^T)^{\alpha+\beta},
> > > \quad
> > > D_3(P^S) = \frac{1}{\alpha(\alpha+\beta)} \sum_j (p_j^S)^{\alpha+\beta}.
> > > $$
> > >
> > >
> > > The first term $D_1 (P^T , P^S)$  couples the student and the teacher probabilities, and taking its derivative with respect to the student probabilities gives
> > > $$\frac{\partial D_1^{(\alpha,\beta)}}{\partial p_j^S}
> > > = - \frac{1}{\alpha}(p_j^T)^\alpha (p_j^S)^{\beta-1}
> > > $$
> > >
> > > Thus, the gradient is proportional to $(p_j^T)^\alpha$. If $p_k ^{\ast T} $ is a top-K probability ($k \in [K]$) and $p_i ^T $ is a tail probability ($i \notin [K]$) , the ratio of their gradient is $\left(\frac{p_k ^{\ast T}}{p_i^T}\right)^\alpha.$ Since $p_k ^{\ast T} \gg p_i^T $,
> > > $\left(\frac{p_k ^{\ast T}}{p_i^T}\right)^\alpha $ is strictly greater than 1 for $\alpha >0$, which means the gradients will be dominated by the top-K probabilities. Top-K dominance vanishes only at the reverse KL limit ($\alpha \rightarrow 0$), where the gradient becomes independent of the teacher probabilities entirely. The additional terms in the divergence do not remove this effect, as $D_2(P^T)$ is independent of $P^S$, and $D_3(P^S)$ does not depend on the teacher distribution.
> > >
> > > This confirms that top-K dominance is not unique to forward KL. TAD provides a principled solution for the forward KL case, where the log-ratio structure admits exact decoupling (Eq. 2). Extending similar corrections to the $\alpha$-$\beta$ family is a promising direction for future work

---

### Decision · Program_Chairs · 2026-04-30

**Decision:**

Accept (regular)

**Comment:**

This paper addresses the issue of a teacher's high-probability tokens dominating the distillation process when training a student. The key remedy is to reweight the teacher-student KL-divergence differently between the top-K and remaining tokens. Additionally, certain techniques are proposed to maintain stability (e.g., sequence-level normalization) and the overall method is shown to be competitive in computational performance with standard distillation. Experiments are given to demonstrate the gains of the approach.

The reviewers generally perceive the work favorably. They find the method simple and practical. Several suggestions made during the discussions could help convey the key contributions better. First, framing the computational argument better (e.g., offline vs. on-policy, scaling behavior) could help the reader better position the role and contributions of the paper (e.g., vs. DistiLLM). Second, more directly addressing the causal link between the tail-awareness and performance gains (such as through ablation studies) would help dispel some of the pushback regarding comparing to other forms of divergences and the question of whether the gains are a result of the tail adjustment or other factors, such as the normalization changes that the approach performs. Lastly, the experimental scope can also be expanded, to show that merit persists across dataset and architecture scales (the authors provide more in the rebuttal, and these should be included.)

Overall, the contributions of the paper are significant and timely and would be of interest to the community. The implementation and experiments look robust, if somewhat limited, and the paper would mostly benefit from better positioning/comparing the method and establishing a more rigorous connection between its success and the tail-based theory.